



# The optical characteristics and sources of chromophoric dissolved organic matter (CDOM) in seasonal snow of northwestern China

Yue Zhou[1], Hui Wen[1], Jun Liu[1], Wei Pu[1], Qingcai Chen[2,3], and Xin Wang[1]

5    [1] Key Laboratory for Semi-Arid Climate Change of the Ministry of Education, College of Atmospheric Sciences, Lanzhou University, Lanzhou 730000, China

[2] School of Environmental Science and Engineering, Shaanxi University of Science and Technology, Xi'an 710021, China

[3] Graduate School of Environmental Studies, Nagoya University, Nagoya 464-8601,

10   Japan

*Correspondence to:* Qingcai Chen (chenqingcai666@163.com) and Xin Wang (wxin@lzu.edu.cn).



**Abstract.**

Chromophoric dissolved organic matter (CDOM) plays an important role in the global carbon cycle and energy budget. A field campaign was conducted across northwestern China from January to February 2012, and surface seasonal snow samples were collected at 39 sites in Xinjiang and Qinghai provinces. Light-absorption measurements, fluorescence measurements and chemical analysis were conducted to investigate the optical properties and potential sources of CDOM in seasonal snow. The abundance of CDOM (the absorption coefficient at 280 nm, $a_{280}$) and the spectral slope from 275 to 295 nm ($S_{275-295}$) ranged from 0.15-10.57 $m^{-1}$ and 0.0129-0.0389 $nm^{-1}$, respectively. The highest average $a_{280}$ ($2.30 \pm 0.52$ $m^{-1}$) and lowest average $S_{275-295}$ ($0.0188 \pm 0.0015$ $nm^{-1}$) in Qinghai indicated that the snow CDOM in this region had strongly terrestrial characteristic. Relatively low regional average $a_{280}$ values were found in sites located to the north of the Tianshan Mountains and northwestern Xinjiang along the border of China ($0.93 \pm 0.68$ $m^{-1}$ and $0.80 \pm 0.62$ $m^{-1}$, respectively). Parallel factor analysis (PARAFAC) identified three types of chromophores that were attributed to two humic-like substances (HULIS, C1 and C2) and one protein-like material (C3). C1 was mainly from soil HULIS, while the potential sources of C2 were complex and included soil, microbial activities, anthropogenic pollution and biomass burning. The good relationship between $a_{280}$ and the intensity of C1 ($R^2 = 0.938$, $p<0.001$) indicated that the CDOM abundance in the surface snow across northwestern China was mainly controlled by terrestrial sources. In addition, the regional variations of sources for CDOM in snow were further assessed by the analysis of chemical species (e.g., soluble



ions) and air mass backward trajectories combined with satellite active fire locations.



# 1 Introduction

Dissolved organic matter (DOM) is widely distributed in natural aquatic ecosystems and plays a key role in the global carbon cycle (Massicotte et al., 2017). Chromophoric dissolved organic matter (CDOM), the light-absorbing constituent of DOM (Helms et

al., 2008), can absorb light from ultraviolet to visible (UV-vis) wavelengths (Stedmon et al., 2000). CDOM in aquatic ecosystems originates from the microbial decomposition of plant matter and is released by organisms within the ecosystems (autochthonous), as well as is imported from surrounding terrestrial environments (allochthonous) (Yao et al., 2011; Zhang et al., 2010). Due to its light-absorbing

properties, CDOM is important in biological processes (Seekell et al., 2015; Thrane et al., 2014), photochemical processes (Helms et al., 2013; Vaehaetalo and Wetzel, 2004) and the energy budget (Hill and Zimmerman, 2016; Pegau, 2002) in natural water bodies.

The light-absorbing properties, composition, and sources of CDOM in lakes, rivers,

wetlands, coasts, estuaries and ocean have been studied all over the world (Andrew et al., 2013; Chen and Jaffe, 2014; Coble, 1996; Organelli et al., 2014; Shao et al., 2016; Stedmon et al., 2011; Wang et al., 2014). Massicotte et al. (2017) summarized the distribution of DOM concentrations, CDOM absorption and their relationship based on more than 12000 measurements across rivers, lakes and oceans globally. Their study

showed that the CDOM from wetlands has the highest absorption coefficients at 350 nm ($a_{CDOM}(350)$) with a median value of 87.2 $m^{-1}$, while the median $a_{CDOM}(350)$ of oceans was 0.08 $m^{-1}$ and CDOM absorption decreased from fresh water to oceans. A

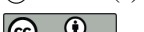

strong logarithmic relationship between DOC concentrations and $a_{CDOM}(350)$ (n =

12808, $R^2$ = 0.92) was also found.

Compared to the aquatic ecosystems, there were far fewer studies evaluating CDOM

in the cryosphere (Barker et al., 2009; Dubnick et al., 2010; Feng et al., 2016; Norman

et al., 2011). However, the global glacier ecosystem is a large organic carbon pool and

exports approximately $1.04 \pm 0.18$ TgC $yr^{-1}$ of DOC into freshwater and marine

ecosystems (Hood et al., 2015). Dubnick et al. (2010) separated the CDOM of ice

samples from several glacial environments into five components, namely, four protein-

like components and one humic-like component, using fluorescence spectrometry.

According to the results of a field experiment conducted in Canada, biological

fluorophores represented the main species of CDOM in subglacially routed meltwater

(Barker et al., 2009). Feng et al. (2016) found that the CDOM in cryoconite samples in

Tibetan Plateau glaciers was mainly derived from microbial activity using light

absorption and fluorescence measurements. In general, CDOM in glacial environments

shows high bioavailability (Hood et al., 2015).

However, to the best of our knowledge, there is no study focusing on the CDOM in

seasonal snow in midlatitude regions. As the largest component of the terrestrial

cryosphere (Brutel-Vuilmet et al., 2013), seasonal snow can cover up to 40% of Earth's

surface during the Northern Hemisphere winter (Hall et al., 1995). For its ecological

functions, snowfall is an important carbon and nutrient input for land ecosystems

(Mladenov et al., 2012) and a crucial freshwater reservoir (Jones, 1999), especially for

barren regions (e.g., northwestern China). In addition, snowpack is also an active field





for photochemical (Beine et al., 2011; Domine et al., 2013) and biological processes
(Liu et al., 2009; Lutz et al., 2016). Unlike aquatic environments, the high surface
albedo is the most obvious physical property of snow (IPCC, 2013). Once light-
absorbing impurities are deposited on the snow surface, the albedo can be significantly

reduced, and the regional and global climate are further affected (Hadley and
Kirchstetter, 2012). Several field campaigns covering the Arctic, Russia, North America
and northern China have been conducted to measure insoluble light-absorbing particles
(ILAPs) in seasonal snow (Doherty et al., 2010, 2014, 2015; Huang et al., 2011; Pu et
al., 2017; Wang et al., 2013, 2015, 2017; Warren and Wiscombe, 1980; Ye et al., 2012;

Zhou et al., 2017a). However, these studies neglected CDOM, which is also an effective
light absorber, whether in the atmosphere (i.e., brown carbon, BrC) or water bodies.
However, Dang and Hegg (2014) suggested that the absorption coefficients of soluble
chromophores in Arctic snow are lower than those of black carbon (BC) by several
orders of magnitudes at visible wavelengths. Compared to the Arctic, the results might

be different in seasonal snow across northwestern China due to the much higher level
of anthropogenic activity and dry deposition of local soil there (Pu et al., 2017).

  UV-vis absorption spectral analysis is a rapid and effective method of characterizing
the optical properties and sources of CDOM. The abundance of CDOM is usually
described by the light absorption of DOM at certain wavelengths within the UV band,

for instance, 280 nm (Zhang et al., 2011). The absorption spectrum of CDOM decreases
approximately exponentially with increasing wavelength (Helms et al., 2008), and the
spectral slope is used to describe the wavelength dependence of absorption



(Twardowski et al., 2004). Helms et al. (2008) used the spectral slope between 275 and 295 nm ($S_{275\text{-}295}$) and the ratio of the spectral slopes ($S_R$) of two narrow bands (275-295 nm and 350-440 nm) to investigate the molecular weight and sources of CDOM (terrestrial or marine origin). Then, $S_{275\text{-}295}$ was used as a tracer of terrigenous DOC

import to marine environments and applied to the Arctic Ocean by remote sensing (Fichot and Benner, 2012; Fichot et al., 2013). Fichot and Benner (2011) presented a method to retrieve the DOM concentration using $S_{275\text{-}295}$ for northern Gulf of Mexico marine water.

  The fluorescence excitation-emission matrix (EEM) offers multiple advantages, such

as high sensitivity, easy sample preparation and non-destructivity to samples (Birdwell and Valsaraj, 2010), and has been widely used to identify the source and composition (humic-like or protein-like) of CDOM in natural waterbodies (Birdwell and Engel, 2010; Coble, 1996; Zhao et al., 2016), rainwater (Zhou et al., 2017b), fog water (Birdwell and Valsaraj, 2010) and aerosols (Chen et al., 2016a, 2016b; Fu et al., 2015).

However, the CDOM in nature is composed of complicated chromophores and shows overlapping signals with EEMs. To precisely extract the useful information from the very large dataset of EEMs, Stedmon et al. (2003) successfully applied parallel factor analysis (PARAFAC) to decompose the EEMs into several independent fluorescent components. Due to the great advantage of PARAFAC in interpreting the results of

EEMs, this has been a "mainstream" approach in recent natural CDOM studies (Murphy et al., 2013). In addition, three fluorescence-derived indices are widely used to identify the potential sources of CDOM. Zsolnay et al. (1999) presented a

humification index (HIX) to describe the relative humification of DOM. The

fluorescence index (FI) is used to identify the sources of DOM from terrestrial or

microbial origins (McKnight et al., 2001), and the biological index (BIX) can be an

indicator of autochthonous productivity (Huguet et al., 2009).

5     In this study, surface snow samples collected across northwestern China were

subjected to UV-vis absorption, fluorescence and chemical analyses. For the first time,

with the aim of presenting an understanding of CDOM in seasonal snow, the

abundances, optical properties and potential sources of CDOM and their spatial

distributions were investigated.

## 10  2 Material and methods

### 2.1 Sample collection

During January to February 2012, snow samples were collected at 7 sites (no. 47- 51a,

51b, 52) in Qinghai and 32 sites (no. 53-84) in Xinjiang, which are located in

northwestern China. The distribution of sample sites, which are numbered

15    chronologically, is shown in Fig. 1. The sample sites were separated into five regions

to investigate the spatial variations of CDOM abundance, optical properties and their

potential sources. One group of sites was in Qinghai, and four groups were in Xinjiang,

following the grouping scheme presented by Pu et al. (2017).

    The sample sites were chosen to be upwind and far enough away from roads, railways,

20    cities and villages to minimize the effects of local pollution. Hence, the collected

samples can be representative of a wide range of areas. Pictures of some of the sample

sites are shown in Fig. 2. At each site, snow samples were collected every 5 cm from

top to bottom, and if there was a melt layer or fresh snow on the top layer, such a sample

was collected individually. A pair of two adjacent vertical profiles of snow were

gathered ("left" and "right") for assessing the variability of the same snowpack and to

5    enhance the accuracy of the measurements. During this campaign, 13 fresh snow

samples that had fallen during the sampling time were collected. In addition, at some

sites, the snow was thin and patchy and the wind was strong; hence, the samples were

gathered from snow drifts. These samples were potentially influenced by the deposition

of local soil (Ye et al., 2012). After being returned to the laboratory in Lanzhou

10    University, all the samples were stored in a freezer at -20°C or lower for subsequent

analysis. More details on the sampling methods have been reported previously (Doherty

et al., 2010; Wang et al., 2013; Ye et al., 2012).

### 2.2 Fluorescence measurement

The snow water samples were first filtrated using 0.22 μm PTFE syringe filters (Jinteng,

15    Tianjin, China), and the filtrates were stored in prebaked glass vials (450 °C for 4 h) at

4 °C until measurement within 24 h. The ultrapure water (18.2 MΩ·cm) filtrated by the

PTFE syringe filters exhibited no clear fluorescence signal.

    The EEMs of surface snow samples were determined by an Aqualog

spectrofluorometer system (Horiba Scientific, NJ, USA) in a 1 cm quartz cell. The

20    scanning ranges were 240 to 600 nm in 5 nm intervals for excitation and 250 to 825 nm

in 4.65 nm (8 pixel) intervals for emission, and the integrating time was 5 s. An





ultrapure water blank was subtracted to remove the water Raman scatter peaks.

The inner filter effect (IFE) of EEM was corrected using the method shown in Kothawala et al. (2013). The fluorescence intensities were calibrated by the Raman peak of the ultrapure water reference at a 350 nm excitation wavelength following the method presented by Lawaetz and Stedmon (2009). The Rayleigh scatter peaks of EEMs were addressed by the EEMscat MATLAB toolbox (version 3) (Bahram et al., 2006) using an interpolation algorithm.

The PARAFAC analysis was performed by the DOMFluor toolbox (version 1.7) in MATLAB (Stedmon and Bro, 2008). In addition, the emission wavelengths longer than 650 nm were removed to eliminate the uncertainty of measurement. According to the analysis of residual error and visual inspection, the three-component PARAFAC model was selected. The residual error decreased distinctly when the components increased from two to three (Fig. S1). The three-component model was also validated by the split-half analysis with the "$S_4C_4T_2$" split scheme (Murphy et al., 2013) (Fig. S2). The fluorescence intensity of the fluorescence component was expressed as $F_{max}$ in Raman unit (RU) (Stedmon and Markager, 2005a). In addition, the fluorescence-derived indices were calculated by Eq. (1-3) (Huguet et al., 2009; McKnight et al., 2001; Zsolnay et al., 1999):

$$FI = I_{370}^{450} / I_{370}^{499} , \tag{1}$$

$$BIX = I_{310}^{379} / I_{310}^{430} , \tag{2}$$

$$HIX = I_{255}^{434-480} / I_{255}^{300-345} , \tag{3}$$

where I is the fluorescence intensity, the subscript denotes the excitation wavelength





and the superscript denotes the corresponding emission wavelength or wavelength range. We note that the wavelengths used in the calculation were changed slightly (1 nm or less) due to different instruments.

### 2.3 UV-vis absorption measurement

The UV-vis absorption spectra of snow samples were derived from 240 to 600 nm in 5 nm intervals while the fluorescence measurements were conducted by an Aqualog spectrofluorometer system (Horiba Scientific, NJ, USA), and an ultrapure water blank was used as a reference. The absorbance of CDOM was assumed to be zero above 550 nm, and the average absorbance between 550-600 nm was subtracted from the whole

spectrum to correct the baseline shifts and scattering effects of the measurement. The absorbances of the samples were converted to absorption coefficients using the following equation:

$$a(\lambda) = \ln(10) \cdot A(\lambda) / L, \qquad (4)$$

where A is the absorbance of the sample, $\lambda$ is the wavelength, L is the path length of

cuvette (0.01 m), and a is the absorption coefficient ($m^{-1}$). The abundance of CDOM is presented by the absorption coefficients at 280 nm ($a_{280}$) (Zhang et al., 2010). The spectral slope between 275-295 nm ($S_{275-295}$) was determined by a linear regression between logarithmic absorbance and wavelengths since the correlation coefficients were higher than that of the exponential fit. If the difference in $S_{275-295}$ between the

linear and exponential regressions was higher than 10%, indicating a high uncertainty for absorption measurement, such data were removed. The absorption Ångström





exponents (AAEs, from 240 to 550 nm) of CDOM were calculated using power-law fit from Eq. (5), as follows:

$$a(\lambda) = K \cdot \lambda^{-AAE}, \tag{5}$$

where a is the absorption coefficient (m$^{-1}$), K is a constant and $\lambda$ is the wavelength.

5    Because the light absorption within the visible wavelengths of some samples were too low to be measured by the spectrometer, 19 of 39 samples were available for the calculation of AAE.

Furthermore, the "left" samples of site 51b and 58 showed abnormal absorption and fluorescence spectra compared to the other samples, and were supposed to be 10    contaminated, and hence, these two samples were not used in the absorption and fluorescence analyses.

**2.4 Soluble ions**

The major soluble ions of surface snow water samples were analyzed with an ion chromatograph (Dionex, Sunnyvale, CA, USA) using an AS11 column for the anions 15    $SO_4^{2-}$, $NO_3^-$, $Cl^-$, and $F^-$ and a CS12 column for the cations $Na^+$, $K^+$, $Ca^{2+}$, $Mg^{2+}$, and $NH_4^+$. The soluble ions showed no obvious differences between filtered and unfiltered samples (Pu et al., 2017). In addition, the concentration of non-sea-salt $K^+$ (nss-$K^+$) was corrected by $Mg^{2+}$ (a sea salt indicator) using the following formula (Tao et al., 2016):

20                          $$\text{nss-}K^+ = K^+ - 0.159Mg^{2+}. \tag{6}$$





## 2.5 Hierarchical cluster analysis

A hierarchical cluster analysis was used to classify the CDOM in snow based on the relative abundances of PARAFAC components. Euclidean distance was used to estimate the distances between samples, and the unweighted average distance was

chosen for the hierarchical cluster analysis by comparing the multiple correlation coefficients for the weighted average, centroid, farthest neighbor, shortest neighbor, weighted center of mass and Ward's distance. A total of four clusters were determined and labeled clusters A-D.

## 2.6 Active fire data and air mass backward trajectories

Active fire data (MCD14DL) from Moderate Resolution Imaging Spectroradiometer (MODIS) Collection 6 were used to capture potential fire source distributions. The data are available online: http://earthdata.nasa.gov/firms. To further analyze the potential sources of CDOM, 72-h air mass backward trajectories were conducted by the HYbrid Single-Particle Lagrangian Integrated Trajectory (HYSPLIT,

http://ready.arl.noaa.gov/HYSPLIT.php) model (version 4). The model was run at 500 m above ground level four times a day for a period of 30 days preceding the sampling date at a given site.

## 3 Results and discussion

### 3.1 The absorption characteristics of CDOM ($a_{280}$, $S_{275-295}$ and AAE)

The distributions of $a_{280}$ and $S_{275-295}$ are shown in Fig. 3 and the corresponding

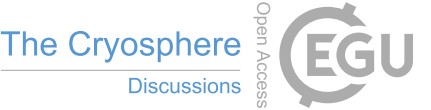



values are summarized in Table 1. $a_{280}$ ranged widely from 0.15 to 10.57 m$^{-1}$ with an

average of 1.69±1.80 m$^{-1}$. The highest value appeared at site 67 (10.57 m$^{-1}$), followed

by sites 53, 79 and 47 (5.25 m$^{-1}$, 3.13 m$^{-1}$ and 3.11 m$^{-1}$, respectively). Most of these

samples were collected from the snow drifts under strong wind. These values were

higher than the $a_{280}$ of snow, ice and cryoconite CDOM on the Tibetan Plateau

(typically lower than 2.0 m$^{-1}$; Feng et al., 2016, 2017). The lowest value was found at

site 66 (0.15 m$^{-1}$), followed by sites 70, 82, 73 and 83 (0.21 m$^{-1}$, 0.23 m$^{-1}$ 0.30 m$^{-1}$ and

0.31 m$^{-1}$, respectively), and these values were compared to the absorption of soluble

light-absorbing species in snow at Alaska with typical values of 0.1-0.15 m$^{-1}$ at 250 nm

(Beine et al., 2011). Some of these samples comprised freshly fallen snow and some

were collected at remote sites in northern Xinjiang, which were far from pollution

sources (Pu et al., 2017). The values of $S_{275-295}$ ranged from 0.0129 to 0.0389 nm$^{-1}$

with an average of 0.0243±0.0073 nm$^{-1}$. Hansen et al. (2016) summarized the values

of $S_{275-295}$ for oceanic and terrestrial systems, and the values range from 0.020-0.030

nm$^{-1}$ for ocean, 0.010-0.020 nm$^{-1}$ for coastal water, and 0.012-0.023 nm$^{-1}$ for terrestrial

river systems. The wide range of $S_{275-295}$ indicates complex sources of CDOM in

seasonal snow across northwestern China. Furthermore, as shown in Fig. 4, $S_{275-295}$

decreases logarithmically with increasing $a_{280}$ ($R^2 = 0.586$, p<0.001). As presented by

Helms et al. (2008), lower $S_{275-295}$ values correspond to terrestrial and higher

molecular weight CDOM. Therefore, the higher $a_{280}$ values of our samples may be

due to greater inputs of terrestrially derived CDOM with high molecular weights, such

as HULIS (Zhou et al., 2013). Similar negative relationships between absorption

coefficients and $S_{275-295}$ was also found in estuarine (Asmala et al., 2012) and marine

water (Zhou et al., 2013). The AAEs of 19 CDOM samples are also shown in Table 1.

The values ranged from 4.41–8.91 with an average of $5.55 \pm 1.11$. This value is

comparable with the average AAE of HULIS (6.11, from 300 to 550 nm), the major

CDOM species in snow (Beine et al., 2011), extracted from Alaskan snow (Voisin et al.,

2012).

    The detailed results of each region are discussed below. Region 1 (sites 47-52) is

located in the eastern Tibetan Plateau in Qinghai province, which is typically higher

than 4000 m above sea level. In this region, the snowpack was usually patchy and thin

(Fig. 2a). During windy time, local soil can be blown and deposited on the snow surface,

this had been observed by previous studies (Pu et al., 2017; Ye et al., 2012), and the

filters for samples in this region were in yellow color due to high loading of soil. The

average $a_{280}$ was highest among all five regions ($2.30 \pm 0.52$ m$^{-1}$), and the $S_{275-295}$ fell

in the range of 0.0170–0.0212 nm$^{-1}$ (mean: $0.0188 \pm 0.0015$ nm$^{-1}$). The lowest regional

average and slight variation of $S_{275-295}$ clearly indicates the dominant contribution of

a terrestrial source (e.g., local soil) to snow CDOM (Fichot and Benner, 2012; Helms

et al., 2008).

    Region 2 (sites 53-59, 61, and 79) is along the Tianshan Mountains. Within this

region, some sites were farmland (e.g., sites 55 and 56) (Fig. 2c), which can be

influenced by local agriculture activities (e.g., crop and grass burning). The snow at

some other sites (e.g., sites 53, 57 and 79) was patchy (Fig. 2b), and blowing soil have

been an important source of CDOM. Thus, the $a_{280}$ values of these sites (1.66-5.25 m$^{-}$





$^{-1}$) were higher than those at the other sites (0.41-0.54 m$^{-1}$) with high snow coverage

rates, new fallen snow or long distances from human activities (high altitude) (e.g., sites

54, 58 and 59). Overall, region 2 showed a high average $a_{280}$ (2.00±1.50 m$^{-1}$), and the

average $S_{275-295}$ was 0.0229±0.0073 nm$^{-1}$.

5    Region 3 (sites 60, 62, 63 and 80-84) is located to the north of the Tianshan

Mountains and close to the industrial city belt in central Xinjiang. Hence, human

activities may dominate the contribution of CDOM in snow in this region. However,

the $a_{280}$ values were mostly less than 1.0 m$^{-1}$ except for at sites 60 and 84, and the

average value was low (0.93±0.68 m$^{-1}$). Because samples of these sites were almost

10   new fallen snow, the deposition of pollutants to the snowpack can be quite slight. Sites

60 and 84 were both close to industrial cities (Fig. 1 in Pu et al., 2017), and the

transportation and deposition of anthropogenic pollutants may be responsible for the

high $a_{280}$ (2.39 m$^{-1}$ and 1.65 m$^{-1}$, respectively). The average $S_{275-295}$ was 0.0218±

0.0057 nm$^{-1}$ and was comparable to that in region 2.

15   Region 4 (sites 64-71) is located in northwestern Xinjiang. In this region, the snow

depth increased from south to north. At sites 64-67, the snow was thin due to the low

snowfall in sample year (Ye et al., 2012). The maximal and minimal $a_{280}$ values of the

entire campaign, 10.57 m$^{-1}$ (site 67, snow drift, Fig. 2e) and 0.15 m$^{-1}$ (site 66, new snow),

were found in this area. Sites 68-71, which were the northmost sites during this

20   campaign, were far from industrial areas, and the snow depth and coverage rate were

high, which led to low $a_{280}$ values. Generally, the $a_{280}$ was in the range of 0.5-2.0 m$^{-}$

$^{1}$ with an average of 0.80±0.62 m$^{-1}$ (excluded site 67), which was the lowest compared

to the values in the other four regions. The mean value of $S_{275-295}$ (0.0255±0.0060 nm$^{-1}$) was higher than that in regions 2 and 3. The low $S_{275-295}$ of site 67 (0.0169 nm$^{-1}$) suggested a strongly terrestrial input of CDOM.

Region 5 (sites 72-78) is located in northeastern Xinjiang along the border of China.
The average value of $a_{280}$ was 1.17±0.63 m$^{-1}$, which was intermediate among five regions. However, the $S_{275-295}$ was typically higher than 0.0300 nm$^{-1}$ with an average of $0.0324 \pm 0.0060$ nm$^{-1}$. These values indicated that non-terrestrial sources were dominant for snow CDOM in this region.

## 3.2 The fluorescence characteristics of CDOM

### 3.2.1 PARAFAC components

The EEMs of snow water samples were analyzed by PARAFAC, and three separate fluorescent components were identified (Fig. 5a-c). Figure 5d-f present the corresponding excitation and emission loading spectra of each component. The excitation/emission (Ex/Em) wavelengths of each component's fluorescence peaks are
summarized in Table 2. The spectral characteristics of three components in this study are similar to the fluorophores identified in aquatic environments in previous studies (Stedmon and Markager, 2005a; Stedmon et al., 2003; Zhang et al., 2010, 2011).

C1 showed a primary peak at <240/453 nm for Ex/Em, which was similar to the component 1 reported by Stedmon and Markager (2005a). This kind of fluorophore
absorbs light mainly in the UVC band and shows a broad emission peak, and is usually identified as terrestrial CDOM (Stedmon et al., 2003). However, in our study, a

secondary peak was also presented (Ex/Em = 305/453 nm). The appearance of an

excitation peak at longer wavelengths may indicate that C1 is more aromatic and has

higher molecular weight (Coble et al., 1998). C1 also resembled another terrestrial

fluorophore, namely, component 4 in Stedmon and Markager (2005a) (Ex/Em = <250,

360/440), which has been widely found in nature fresh water environments and even

water-extracted organic matter in aerosols (Chen et al., 2016b; Mladenov et al., 2011;

Zhang et al., 2009; Zhao et al., 2016). C1 in our study showed blue-shifted excitation

wavelengths, which likely corresponded to a lower molecular weight (Coble et al.,

1998).

C2 had a primary (secondary) peak at <240(300)/393 nm (Ex/Em), namely, peak M,

that was first measured in the oceanic system by Coble (1996). Hence, this component

is usually identified as marine HULIS. However, the marine sources of CDOM can be

nearly ignored in this study because the sample sites were several thousand kilometers

away from the ocean. Stedmon et al. (2003) found a similar fluorophore (component 4

therein) that showed terrestrial source in a terrestrially dominated estuary region. The

subsequent studies suggested that the C2-like component is also linked to microbial

activity and phytoplankton degradation in natural aquatic systems (Yamashita et al.,

2008; Zhang et al., 2009) or DOM in wastewater from anthropogenic sources (Stedmon

and Markager, 2005a).

C3 is a typical fluorophore that is categorized as tyrosine-like CDOM and that

exhibits Ex/Em pairs of <240(270)/315 nm. C3 reflects autochthonously labile CDOM

produced by biological processes (Stedmon et al., 2003) and has been commonly

reported in previous studies of natural water bodies and the water extraction of aerosols

(Chen et al., 2016b; Murphy et al., 2008; Stedmon and Markager, 2005b).

To analyze the mutual relationships between three components, linear correlation

analyses were conducted (Fig. 6). The $F_{max}$ values of C3 at site 67 were much higher

than those of any other sample (shown as red markers in Fig. S3), which can strongly

influence the results of the correlation analysis. When excluding the data of site 67, the

$R^2$ between $F_{max}$(C1) and $F_{max}$(C3) fell from 0.316 to 0.082, and the linear relationship

became nonsignificant (Fig. S3). Therefore, we chose to use the dataset that excludes

site 67 in the analysis, and the results are shown below. $F_{max}$(C1) and $F_{max}$(C2) exhibited

a significant and positive correlation ($R^2 = 0.332$, $p<0.001$); however, this value was

much lower than that in previous studies of natural water, for instance, $R^2 = 0.63$ for

inland lakes (Zhao et al., 2016) and $R^2 = 0.88$ for inland rivers (Zhang et al., 2011). This

result indicated that two humic-like components might only share partially common

sources. The relationships between the two humic-like components and the protein-like

component are quite different. Not surprisingly, $F_{max}$(C1) and $F_{max}$(C3) showed no

correlation ($R^2 = 0.082$, $p>0.05$), while a significant linear relationship ($R^2 = 0.364$,

$p<0.001$) was found between $F_{max}$(C2) and $F_{max}$(C3), which implied a potential

microbial source for C2, consistent with the finding of Yamashita et al. (2008).

### 3.2.2 Regional variation in PARAFAC components

Figure 7 shows the relative intensities of the three fluorescence components at each site,

and Table 3 exhibits the regional average of the relative intensity and $F_{max}$ for each

component. Overall, C2 was the most intense fluorophore and accounted for 42% on average of the total fluorescence intensity of all samples, followed by C3 (38% on average) and C1 (20% on average). Compared to glacial snow and ice samples, which were dominated by protein-like substances (Dubnick et al., 2010; Feng et al., 2016), the

seasonal snow samples in this study showed fewer microbial characteristics.

In Qinghai (region 1), the most obvious feature was that C1 accounted for approximately 35% of the total fluorescence intensity. This value was significantly higher than the other regions. As mentioned in Sect. 3.1, the snowpack of these sites was strongly influenced by the local soil, hence clearly implying that C1 was mainly

from the soil HULIS. This finding confirmed the constantly terrestrial source of the C1-like fluorophore, regardless of whether the sample was collected from natural water bodies, aerosol water extraction or snow. The relative contribution of C3 (the protein-like component) in region 1 was much lower than that in other regions. This result was mainly due to the high $F_{max}(C1)$ in region 1 since the regional variation of $F_{max}(C3)$ was

slight (Table 3).

In Xinjiang, the relative intensity of C1 varied by region, while C2 and C3 accounted for roughly equal contributions. In region 2, the relative contribution of C1 was high (25% on average), which implied a substantial amount of CDOM originating from local soil. In region 3, where most of the samples were new fallen snow (7 of 8 sites), the

CDOM showed much fewer terrestrial characteristics as demonstrated by the lowest relative intensity of C1 (9% on average). The great difference between relative contributions of C1 and C2 in this region indicated that the sources of these two humic-





like components were different. In regions 4 and 5, the contributions of C1 (both were approximately 17%) were nearly double that in region 3. As the snow samples collected in northern Xinjiang were more aged than those in region 3, the deposition of soil was more significant, which led to a higher C1 fraction.

At site 54 and 82, the fractional intensity of C3 exceeded 70%. This value was approximately twofold higher than the average of all samples. This result can be explained by two possible reasons, (1) lower inputs of C1 and C2 were present at these sites, and (2) much greater microbial activities were available in the snowpack at sites 54 and 82. We found algal communities near the sample sites (Fig. S4), providing

evidence for the latter reason.

At site 67, the fluorescence intensities of C1, C2 and C3 were all highest among all samples (0.30 RU, 0.39 RU and 0.38 RU, respectively), especially for C3. The average intensity of C3 for samples excluding site 67 was 0.10 RU, with a low standard deviation of 0.02 RU, and this value was approximately one-fourth of that at site 67.

Therefore, rather than owing to microbial activity alone, the extremely high $F_{max}$(C3) of site 67 may be due to other sources, for instance, some organic compounds released from diesel combustion may show similar spectra (Mladenov et al., 2011).

To assess the similarities and differences between samples from different regions, a hierarchical cluster analysis based on the relative intensities of three fluorescence

components was conducted (Fig. 8). The snow samples were separated into four clusters (clusters A-D) (Fig. S5). Samples classified into clusters A and B were dominant. The high relative contribution of C1, which was 34% on average, was the most remarkable

feature of cluster A and led to a low relative abundance of C3 (26% on average) (Table
S1). All samples in region 1 and most samples in region 2 were assigned to cluster A.
For cluster B, the contribution of C1 was low (13% on average), and the abundance of
C3 (47% on average) was slightly higher than that of C2 (40% on average). For sites in
northern Xinjiang (regions 4 and 5), most samples were classified into cluster B. The
samples assigned to cluster C, including those of sites 60, 62, 69, 72, 76 and 84, showed
dominant contribution of C2 (57% on average). Half of these samples were found in
region 3, and the others were dispersed in regions 4 and 5. Pu et al. (2017) analyzed the
concentrations and sources of ILAPs in snow in the same field campaign. They found
that the dominant source of ILAPs at these sites was industrial pollution. This result
indicates that the humic-like fluorophore C2 may partly originate from anthropogenic
pollution. Cluster D contained only two samples from sites 54 and 82. The difference
between cluster D and others was an extremely high relative contribution of protein-
like component C3 (73% on average), which indicated the high bioavailability of snow
CDOM there.

### 3.2.3 Fluorescence-derived indices

The values of the three established fluorescence-derived indices for surface snow
samples are shown in Fig. 9. The regional average values are shown in Table 4. The
HIX of samples in this study fell into the range of 0.16-3.20 with an average of 1.21±
0.78. The highest average HIX appeared in region 1 (2.21±0.42), which was highly
consistent with the low $S_{275-295}$ and high relative intensity of C1 and further

demonstrated the large terrestrial contribution in this region. The lowest average HIX was found in region 3 (0.62±0.37 on average), which suggests that the CDOM was fresh. This finding is easily explained as the snow samples in this region were nearly from new fallen snow. Compared to the HIX of other types of samples (Table 5), the

HIX of snow across northwestern China was higher than that of spring water (Birdwell and Engel, 2010); comparable to cryoconite in glaciers from the Tibetan Plateau (Feng et al., 2016), inland lakes (Zhang et al., 2010), cave water (Birdwell and Engel, 2010) and north Pacific Ocean water (Helms et al., 2013); and was lower than estuarine water (Huguet et al., 2009), fog water (Birdwell and Valsaraj, 2010), groundwater (Huang et

al., 2015), water extraction of alpine aerosol (Xie et al., 2016) and urban aerosol (Mladenov et al., 2011).

According to McKnight et al. (2001) and Huguet et al. (2009), the values of BIX > 1.0 or FI > 1.9 indicate microbially derived DOM. The values of BIX and FI for the snow samples were typically below 1.0 and 1.9, respectively, implying unremarkably

autochthonous characteristics. The regional distributions of BIX and FI corresponded with that of HIX. The samples with highest average BIX and FI were in region 3 (0.93 ±0.25 and 1.60±0.15, respectively), and the samples in region 1 exhibited the lowest average values of BIX and FI (0.49±0.05 and 1.29±0.05, respectively). The BIX and FI of different types of samples changed little, and the only exception was the FI of

cryoconite in glaciers from the Tibetan Plateau (3.24, Feng et al., 2016), which was approximately twice as high as those of the other samples.



### 3.3 Correlation between light absorption and fluorescence properties.

The relationships between the $a_{280}$ and $F_{max}$ of each component and the fluorescence-derived indices are shown in Fig. 10. Similar to Fig. 6, we used the dataset excluding site 67 in the analysis, and the comparisons with the results using the entire dataset are

shown in Fig. S6. The $F_{max}(C1)$ exhibited a strong and positive linear relationship with $a_{280}$ ($R^2 = 0.938$, $p<0.001$), while the correlation coefficients between $a_{280}$ and $F_{max}(C2)$ ($R^2 = 0.464$, $p<0.001$), $F_{max}(C3)$ ($R^2 = 0.152$, $p<0.05$) were much lower. These results suggested that the variation of CDOM in northwestern China seasonal snow was mainly controlled by terrestrially derived HULIS. This finding is different from those

results found for inland lakes and rivers (Zhang et al., 2010, 2011; Zhao et al., 2016), in which the correlations between CDOM absorption and C1- and C2-like fluorophores were comparable and both strong.

The relationships between $a_{280}$ and the three fluorescence-derived indices are also shown (Fig. 10d-f). The results showed considerable similarity to that between $a_{280}$

and $F_{max}$. $a_{280}$ and HIX exhibited a significant and positive correlation ($R^2 = 0.828$, $p<0.001$). This result suggested that the CDOM abundance in northwestern China seasonal snow increases with the degree of humification. Negative log-log correlations were found between $a_{280}$ and BIX ($R^2 = 0.269$, $p<0.001$), FI ($R^2 = 0.547$, $p<0.001$), which indicates that a higher CDOM abundance corresponds to higher terrestrial inputs.

The AAE of CDOM also showed a significant correlation with the fluorescence-derived indices (Fig. S7). HIX was negatively and linearly correlated with AAE ($R^2 = 0.396$, $p<0.005$), indicating that the AAE can decrease with the increasing humification

degree of snow CDOM, which can be characterized by the higher aromaticity (Bayer et al., 2000). This result was consistent with a study of BrC in aerosol from the Los Angeles Basin (Zhang et al., 2013). They suggested that compared to water-soluble BrC, the lower AAE of methanol-extracted BrC can be explained by the higher aromaticity

of insoluble organic components. In contrast, positive correlations were found between AAE and BIX ($R^2 = 0.814$, $p<0.001$) and FI ($R^2 = 0.463$, $p<0.005$), which implied that the microbial-derived CDOM has a higher AAE than terrestrial CDOM.

### 3.4 Source distributions of CDOM

As discussed in Sect. 3.2.1, C2 shared partially common sources with C1 and C3, which

represented soil and microbial origins, respectively. Furthermore, the $F_{max}$(C2) and $F_{max}$(C3) varied slightly among regions, while the variation of $F_{max}$(C1) was dramatic (Table 3). This phenomenon clearly reflected that the HULIS fluorophore C2 may originate from sources besides soil and microbial activity. To further analyze the potential sources of PARAFAC components, the correlation coefficients between three

major ions and $F_{max}$ of fluorescence components were calculated. The results are shown in Table 6. $F_{max}$(C2) showed a significant and positive correlation with three ions ($p<0.001$). The secondary ions $SO_4^{2-}$ and $NO_3^-$ are commonly considered as the markers of anthropogenic emissions from the burning of fossil fuel, such as oil and coal (Doherty et al., 2014; Oh et al., 2011; Pu et al., 2017), and nss-$K^+$ is widely used as a

tracer of biomass burning (Tao et al., 2016). Therefore, C2 may also originate from anthropogenic pollution and biomass burning; in other words, there were two additional





potential sources of snow CDOM in this study. Since the contribution of microbial-derived C3 to $a_{280}$ was low (Fig. 10c), three major sources of snow CDOM were identified, namely, soil, biomass burning and anthropogenic pollution. The regional variations of CDOM sources are discussed below using analyses of chemical species

and air mass backward trajectories. In addition, the sources of CDOM in snow are also compared with those of particulate light absorption of ILAPs.

In Qinghai (region 1), the backward trajectories to a typical site (site 47, Fig. 11a) were mostly from the west, and very few fire locations were encountered. Combined with the low ratio of $(SO_4^{2-}+NO_3^-)/nss$-$K^+$ (Fig. 12), the CDOM produced by biomass

burning and anthropogenic pollution is negligible in region 1. As mentioned in Sect. 3.1, the snowpack in Qinghai was strongly influenced by local soil, and hence, the soil HULIS was clearly the primary source of CDOM. In region 2, the contribution of soil to CDOM was also significant, which was confirmed by the high relative intensity of C1, as discussed in Sect. 3.2.2. For the results of the backward trajectories, along the

paths of the air masses to site 55 in region 2 (Fig. 11b), the local fire spots due to the agricultural activities could also be important contributors to snow CDOM. Additionally, the value of $(SO_4^{2-}+NO_3^-)/nss$-$K^+$ was also low in region 2, which indicated an insignificant role of anthropogenic pollution. Therefore, in region 2, a mixed source of soil and biomass burning is reasonable. In region 3, the extremely high

ratio of $(SO_4^{2-}+NO_3^-)$ to $nss$-$K^+$ implied a strong contribution of anthropogenic pollution to snow CDOM. Additionally, the mass ratio of $Cl^-$ and $Na^+$ (2.48, Fig. S8) was also significantly higher than that in sea water (1.18, Hara et al., 2004), which

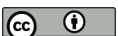


indicated that Cl⁻ might originate from other sources in addition to sea salt, such as coal

combustion (Wang H. L. et al., 2008; Wang Y. et al., 2006), while the values in other

regions were comparable to 1.18. These results, again, confirmed that the CDOM from

anthropogenic pollution was dominant in region 3 but inapparent in other regions. The

backward trajectories also showed consistent results (Fig. 11c). Most of the trajectories

to site 84 came from the northwest and passed through the cities with heavy industry

(e.g., Karamay and Shihezi). Therefore, air pollutants can be transported to the sample

area and deposited in the surface snow. In region 4, the ratios of $(SO_4^{2-}+NO_3^-)$ to nss-$K^+$

were much lower than those in region 3, and the air masses originated from Siberia (Fig.

11d and e), where significant crop and grass burning take place (Hegg et al., 2010),

strongly influenced this region. In region 5, the value of $(SO_4^{2-}+NO_3^-)$/nss-$K^+$ was

comparable with that in region 4, which suggested CDOM from biomass burning rather

than pollution. Although the backward trajectories were more dispersive (Fig. 11f), the

amount of air mass from the Siberia was also respectable which can explain this finding.

Furthermore, the low mass ratios of Cl⁻ and Na⁺ in region 4 and 5 also implied the slight

influence of anthropogenic pollution. Overall, biomass burning was the dominant

source of CDOM in snow in both regions 4 and 5.

Pu et al. (2017) identified the potential sources of particulate light absorption of

ILAPs (denoted as $C_{BC}^{max}$) in snow during the same field campaign, using a positive

matrix factorization (PMF) model. The comparison of $C_{BC}^{max}$ and $a_{280}$ among regions

is shown in Fig. S9. In regions 1 and 5, there were no correlation between $C_{BC}^{max}$ and

$a_{280}$, which indicated entirely different sources of CDOM and particulate absorption of

ILAPs. As reported by Pu et al. (2017), the dominant sources of $C_{BC}^{max}$ were biomass

burning and industrial pollution in regions 1 and 5, respectively, while those of CDOM

were soil and biomass burning in our study. Much higher correlation coefficients were

found in regions 2-4 ($R^2$ = 0.72, 0.95 and 0.81, respectively), especially for region 3,

which implied consistency for sources of CDOM and particulate absorption of ILAPs.

The relative weaker relationship presented in region 2 may have resulted from the

mixed sources of CDOM (soil and biomass burning); however, biomass burning was

the primary source for particulate absorption of ILAPs.

### 3.5 Comparing the light absorption by CDOM and BC

Figure 13 shows the relative contributions of CDOM and BC to light absorption. As

mentioned above, light absorption within visible wavelengths was available for 19

samples. The BC concentrations in surface snow were obtained from Pu et al. (2017),

and the mass absorption coefficient (MAC) at 550 nm and the AAE of BC used in the

calculation were 6.3 $m^2$ $g^{-1}$ and 1.1, respectively (Pu et al., 2017).

The light absorption of CDOM was 0.02-1.17 times that for BC at 400 nm with an

average of 0.34±0.34. At 500 nm, this value decreased quickly to 0.10±0.11 on average

and ranged from 0.005-0.40. The CDOM absorption at 400 nm was comparable to or

even slightly higher than the absorption by BC at sites 50, 52 and 79. This finding is

quite different from the results for Alaskan snow. Dang and Hegg (2014) converted the

CDOM absorption in snow at Barrow, Alaska, reported by Beine et al. (2011), into

equivalent BC mixing ratios of 0.14 ng $g^{-1}$ at 400 nm and 0.07 ng $g^{-1}$ at 550 nm. As



presented by Doherty et al. (2013), the mixing ratio of BC in Barrow snow ranged from 10-30 ng g$^{-1}$. Hence, the absorption of CDOM in Alaskan snow can be safely ignored, but this does not appear reasonable for some sites across northwestern China.

5 Previous studies on impurities in seasonal snow have focused on insoluble particles (e.g., BC, insoluble OC and dust) (Doherty et al., 2010, 2014; Pu et al., 2017;Wang et al., 2013). The above discussion indicated that in some specific areas of northwestern China, the absorption of CDOM in snow was remarkable. What is the common feature of such sites? The average $S_{275-295}$ (0.0187±0.0022) of these 19 sites was lowest compared to the five regional averages, and the average BIX (0.60±0.20) and FI (1.31 10 ±0.09) were lower than those of region 2, in which the influence of local soil was obvious; in addition, the average values of HIX (1.87±0.57) and relative contribution of C1 (30±8%) were higher than those of region 2. These results indicated that the CDOM of these sites was undoubtedly of terrestrial origin (e.g., wind-blown soil). Hence, we suggest that the absorption by CDOM in the snowpack, which is heavily 15 affected by soil, cannot be ignored.

## 4 Conclusions

Surface snow samples was collected across northwestern China from January to February 2012. The $a_{280}$ and $S_{275-295}$ of snow water samples ranged from 0.15-10.57 m$^{-1}$ and 0.0129-0.0389 nm$^{-1}$, respectively. The average value of $a_{280}$ (1.69±1.80 m$^{-1}$) 20 was approximately 10 times higher than that of snow in Alaska (Beine et al., 2011). Samples in Qinghai (region 1) exhibited the highest average $a_{280}$ (2.30±0.52 m$^{-1}$) and

lowest average $S_{275-295}$ (0.0188±0.0015 nm$^{-1}$) resulting from the strong influence of

local soil. Low average $a_{280}$ appeared in central Xinjiang (region 3, 0.93±0.68 m$^{-1}$),

where the samples were nearly collected from new fallen snow, and northwestern

Xinjiang (region 4, 0.80±0.62 m$^{-1}$ when excluded site 67) which was far from industrial

areas. In the Tianshan Mountains (region 2) and northeastern Xinjiang (region 5), the

average $a_{280}$ values were 2.00±1.50 m$^{-1}$ and 1.16±0.63 m$^{-1}$, respectively. For all sites

in Qinghai and some of the sites in Xinjiang (19 of 39 sites), the light absorption of

CDOM cannot be neglected and even was remarkable at some sites compared to that of

BC (0.34±0.34 times relative to BC at 400 nm on average) due to the high contribution

of soil to CDOM in snow. Hence, we suggest that the CDOM absorption at visible

wavelengths at such sites should be taken into consideration in future studies.

Based on PARAFAC, two humic-like fluorophores (C1 and C2) and one protein-like

fluorophore (C3) were identified. C2 showed partially common sources with

terrestrially derived C1 ($R^2$ = 0.332, p<0.001). The positive relationship between

$F_{max}$(C2) and $F_{max}$(C3) ($R^2$ = 0.364, p<0.001) indicated a potential microbial source of

C2. In region 1, the relative intensity of C1 (35%) was much higher than that of the

other regions, and combined with the highest HIX and lowest BIX and FI, indicated a

clearly terrestrial origin of C1. In contrast, the contribution of C1 to total fluorescence

was lowest in region 3 (9%), and the values of fluorescence-derived indices also showed

the consistent results. The high correlations between $a_{280}$ and $F_{max}$(C1) ($R^2$ = 0.938,

p<0.001), HIX ($R^2$ = 0.828, p<0.001) indicated that the CDOM abundance in surface

snow across northwestern China was mainly controlled by the terrestrial sources. A

hierarchical cluster analysis was used to classify samples into four clusters (A-D) on the basis of the relative intensity of three fluorescent components. All samples in region 1 and most samples in region 2 were assigned to cluster A (a high contribution of C1). The number of samples assigned to cluster B (roughly equal contributions of C2 and

C3) and cluster C (a dominant contribution of C2) were nearly even in region 3. For regions 4 and 5, most samples were classified into cluster B. Only two samples were assigned to cluster D due to the dominant contribution of C3.

According to the correlation analysis between three major ions and the $F_{max}$ of PARAFAC components, in addition to soil and microbial activity, C2 exhibited

potential sources of anthropogenic pollution and biomass burning. Furthermore, the spatial variation of CDOM sources was assessed by using regional variations of $(SO_4^{2-}+NO_3^-)/nss-K^+$, $Cl^-/Na^+$ and air mass backward trajectory analysis. In regions 1 and 3-5, the dominant sources were soil, anthropogenic pollution, biomass burning and biomass burning, respectively, while there may be a soil-biomass-burning mixed source

in region 2.

This study investigated the light absorption, fluorescence properties and potential sources of CDOM in seasonal snow across northwestern China. Future studies should focus on the molecular characteristics and the relationships between the optical properties of CDOM in snow. Understanding the structures, chemistry, and sources of

snow CDOM and its effects on the carbon cycle is important.

*Data availability.* All datasets and codes used to produce this study can be obtained by contacting Xin Wang (wxin@lzu.edu.cn). The elevation data used in this study are




available at http://rda.ucar.edu/datasets/ds759.3/#!access.

*Competing interests.* The authors declare that they have no conflict of interest.

*Acknowledgements.* This research was supported by the Foundation for Innovative

Research Groups of the National Natural Science Foundation of China (41521004), the

5    National Natural Science Foundation of China under grant (41522505 and 41775144),

and the Fundamental Research Funds for the Central Universities (lzujbky-2018-k05).

We thank Jinsen Shi of Lanzhou University, Hao Ye of Texas A&M University, and

Rudong Zhang of Nanjing University for their assistance in field sampling.



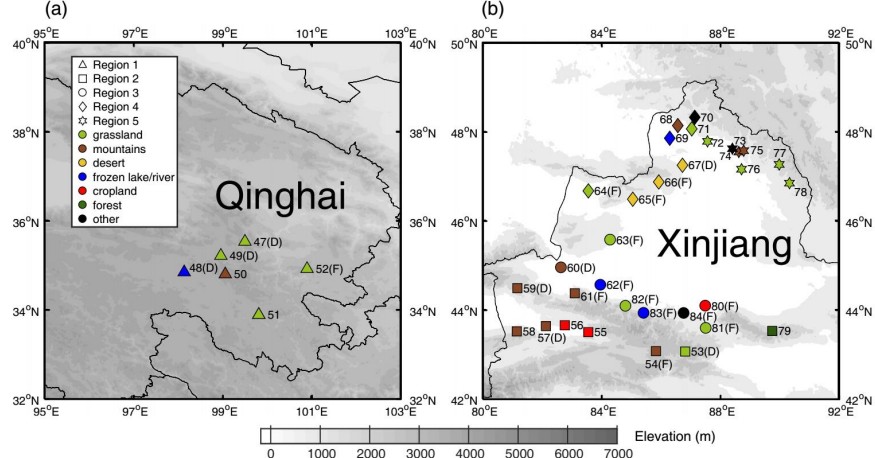

**Figure 1.** Sample site distribution, site numbers and regional groupings in **(a)** Qinghai and **(b)** Xinjiang. Sample areas are divided into five regions indicated by different symbol shapes, and the land cover types of sample sites are represented in different colors, as shown in the legend. The "D" indicates that the sample was collected from a snow drift, and the "F" indicates that the surface sample was fresh snow. The elevation is shown in the contour plot.





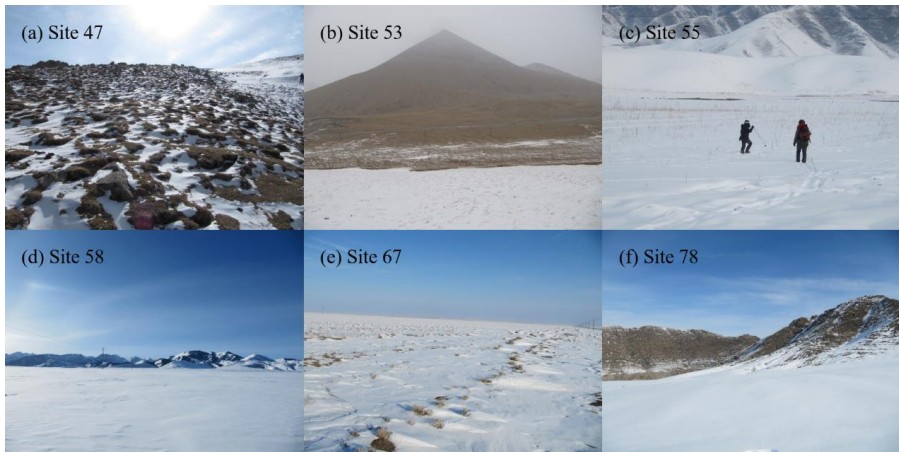

**Figure 2.** Pictures of typical sample sites.





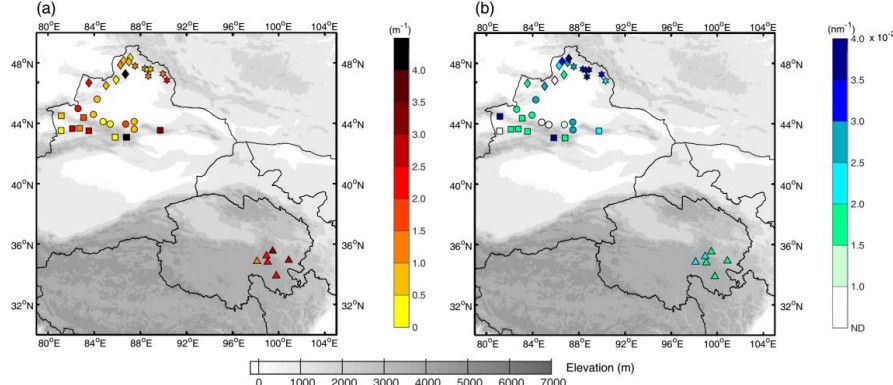

**Figure 3. (a)** $a_{280}$ and **(b)** $S_{275\text{-}295}$ of surface snow at each site. The five regions are

indicated by different symbols (same as Fig. 1).




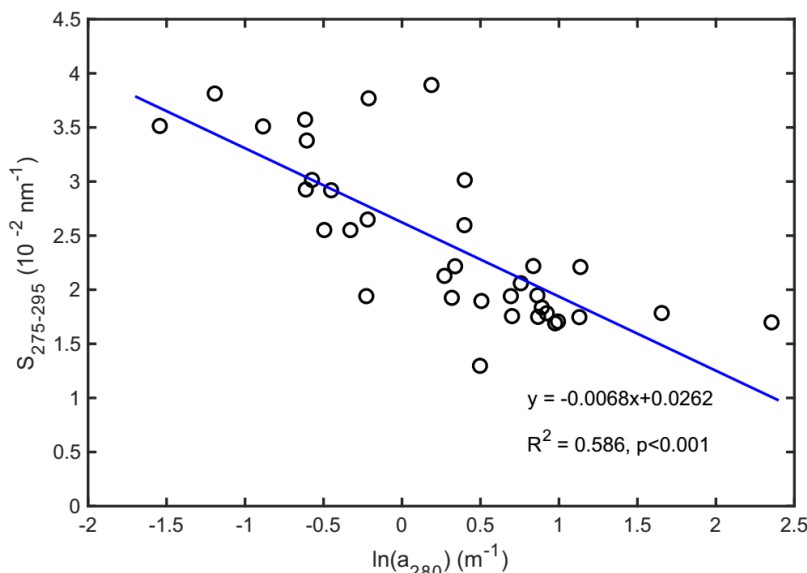

**Figure 4.** Relationship between log-transformed $a_{280}$ and $S_{275\text{-}295}$





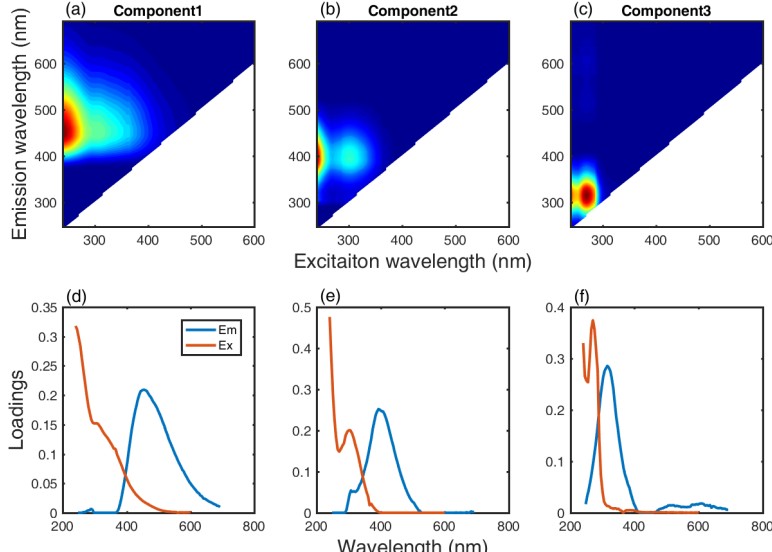

**Figure 5. (a-c)** The EEM components identified by the PARAFAC model and **(d-f)** the corresponding excitation and emission loadings of each component; the orange line indicates the excitation (Ex) loading, and the blue line indicates the emission (Em) loading.





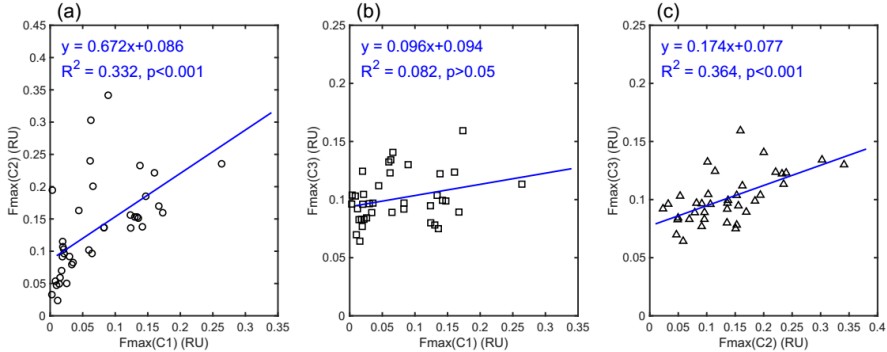

**Figure 6.** The linear relationships between the intensities of **(a)** C1 and C2, **(b)** C1 and

C3, **(c)** C2 and C3.



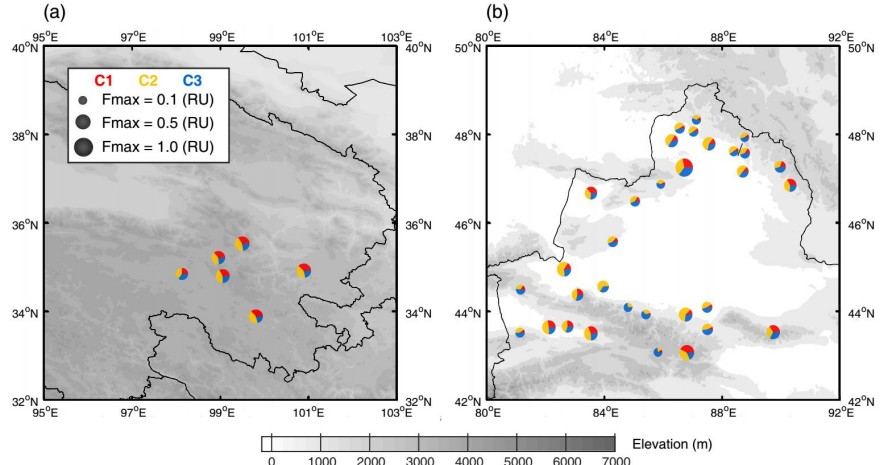

**Figure 7.** Distribution of relative intensities for three fluorescence components in **(a)**

Qinghai and **(b)** Xinjiang. The red, yellow and blue in each pie represent C1, C2 and

C3, respectively, and the size of each pie shows the total fluorescence intensity as a sum

5   of the three components.





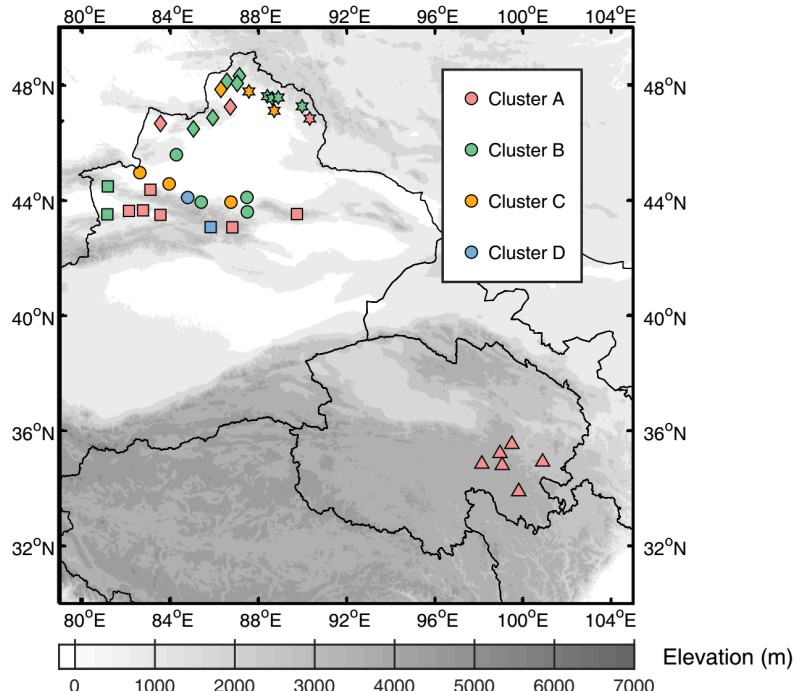

**Figure 8.** Hierarchical cluster analysis based on the relative intensities of the three

PARAFAC components.





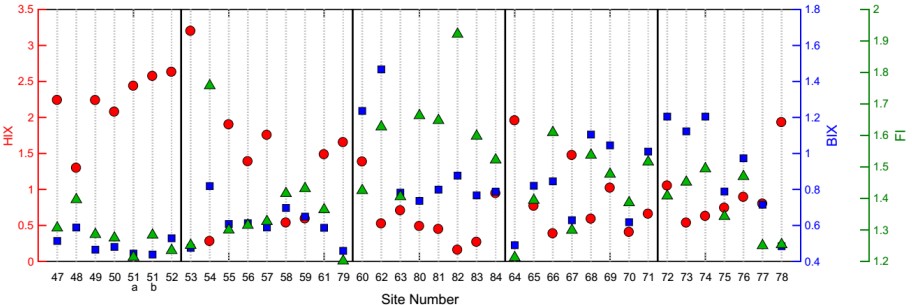

**Figure 9.** HIX (red y-axis and circles), BIX (blue y-axis and squares) and FI (green y-axis and triangles) of surface snow samples. The sample sites in regions 1-5 are separated by black solid lines.





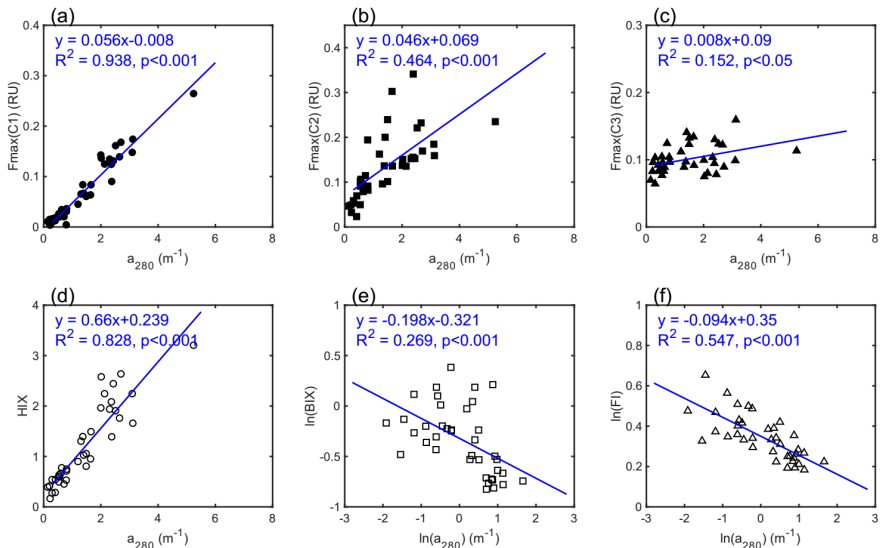

**Figure 10.** The relationships between $a_{280}$ and the intensities of **(a)** C1, **(b)** C2, and **(c)**

C3 and the fluorescence-derived indices **(d)** HIX, **(e)** BIX, and **(f)** FI.



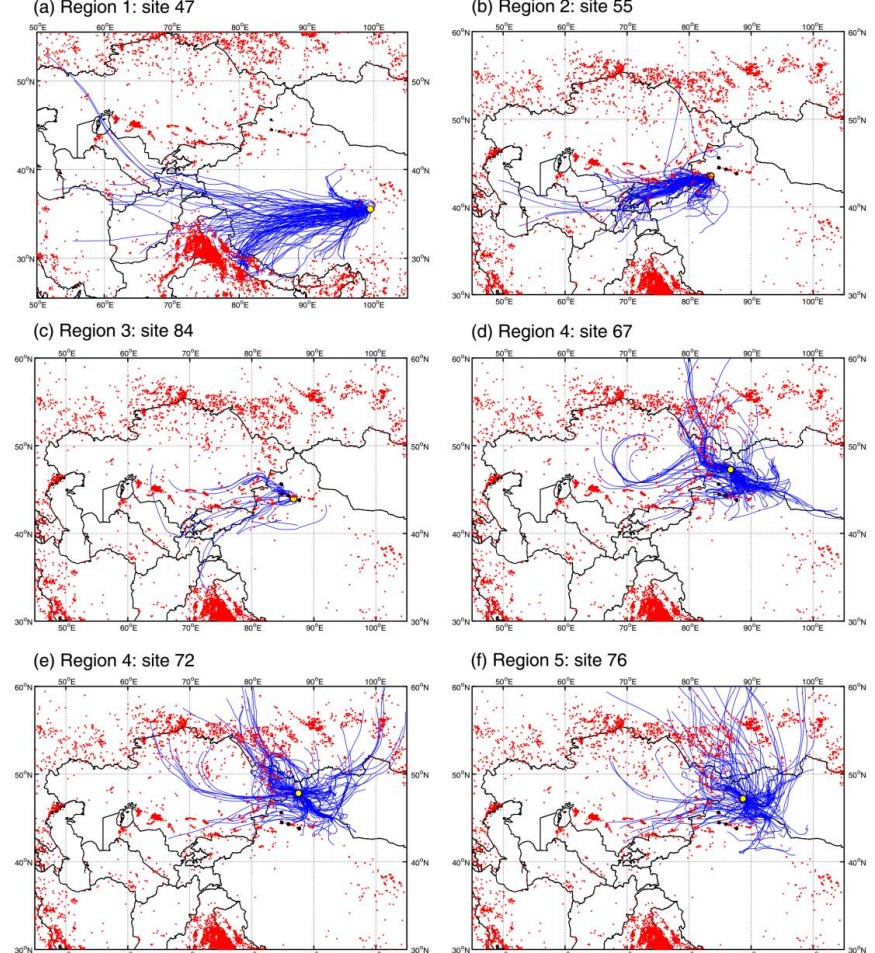

**Figure 11.** 72-h air mass backward trajectories (blue lines) at 500 m above ground level with the initial positions at representative sites (shown as yellow dots) in each region. Trajectories were calculated four times per day for a period of 30 days preceding the sampling date at a given site by HYSPLIT (version 4, NOAA) except for panel (c). Since the snow was fresh at site 84, the trajectories were derived for 5 days preceding the sampling date. The black dots represent the typical industrial cities in Xinjiang, namely, Karamay, Kuytun, Shihezi and Urumqi from west to east. The red dots are MODIS active fire locations.




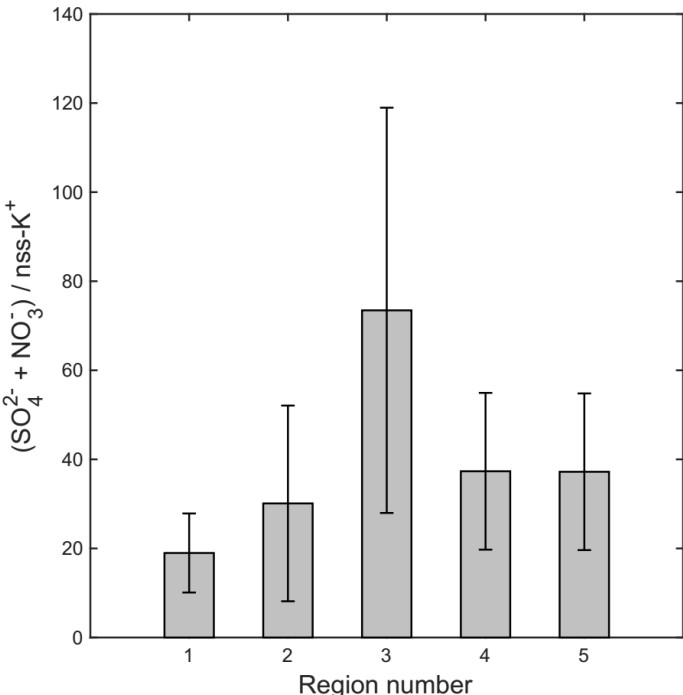

**Figure 12.** The regional averages of the ratio of $(SO_4^{2-}+NO_3^-)$ and $nss\text{-}K^+$.





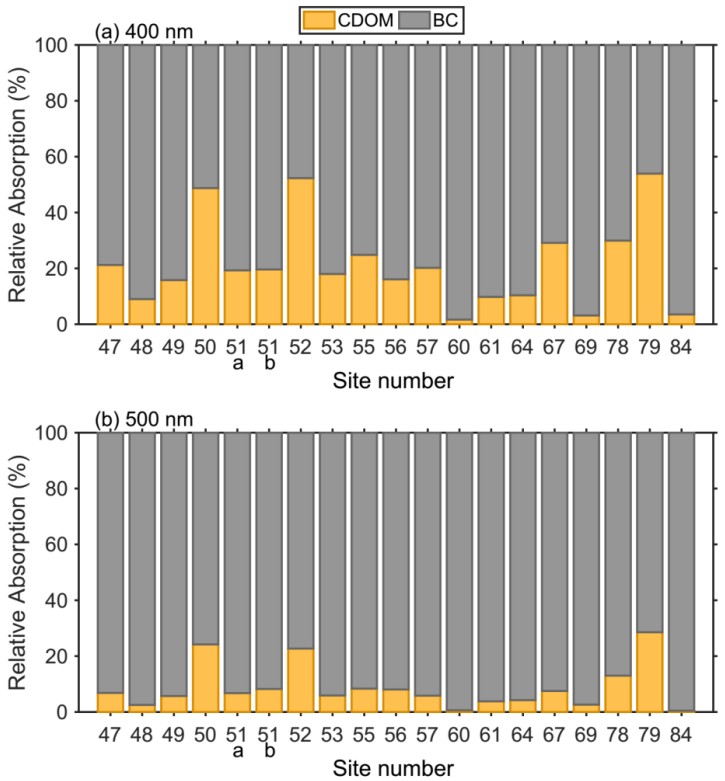

**Figure 13.** The relative absorption contributions of CDOM (yellow bar) and BC (gray

bar) at **(a)** 400 nm and **(b)** 500 nm

**Table 1.** Statistics on absorption and fluorescence parameters for surface snow at each

site. Note: N. A. for no data.

| Site | Lat. (N) | Lon. (E) | $a_{280}$ (m$^{-1}$) | $S_{275-295}$ (nm$^{-1}$) | AAE | HIX | BIX | FI |
|---|---|---|---|---|---|---|---|---|
| 47 | 35.54 | 99.49 | 3.11 | 0.0174 | 4.66 | 2.24 | 0.51 | 1.31 |
| 48 | 34.85 | 98.13 | 1.32 | 0.0212 | 5.12 | 1.30 | 0.59 | 1.40 |
| 49 | 35.22 | 98.95 | 2.14 | 0.0206 | 5.20 | 2.24 | 0.47 | 1.29 |
| 50 | 34.80 | 99.05 | 2.38 | 0.0194 | 4.91 | 2.08 | 0.48 | 1.27 |
| 51a | 33.89 | 99.80 | 2.44 | 0.0183 | 4.87 | 2.44 | 0.44 | 1.21 |
| 51b | 33.89 | 99.80 | 2.02 | 0.0175 | 4.91 | 2.57 | 0.44 | 1.28 |
| 52 | 34.92 | 100.89 | 2.71 | 0.0170 | 4.63 | 2.63 | 0.53 | 1.23 |
| 53 | 43.07 | 86.81 | 5.25 | 0.0178 | 4.53 | 3.20 | 0.48 | 1.25 |
| 54 | 43.08 | 85.82 | 0.41 | 0.0350 | N.A. | 0.28 | 0.82 | 1.76 |
| 55 | 43.51 | 83.54 | 2.52 | 0.0178 | 5.13 | 1.90 | 0.61 | 1.30 |
| 56 | 43.66 | 82.75 | 1.38 | 0.0192 | 5.77 | 1.39 | 0.61 | 1.31 |
| 57 | 43.64 | 82.11 | 2.66 | 0.0168 | 5.31 | 1.75 | 0.59 | 1.33 |
| 58 | 43.52 | 81.13 | 0.42 | N.A. | N.A. | 0.54 | 0.70 | 1.42 |
| 59 | 44.49 | 81.15 | 0.54 | 0.0357 | N.A. | 0.59 | 0.65 | 1.43 |
| 60 | 44.96 | 82.63 | 2.39 | 0.0174 | 8.91 | 1.38 | 1.24 | 1.43 |
| 61 | 44.38 | 83.09 | 1.66 | 0.0189 | 6.06 | 1.49 | 0.59 | 1.36 |
| 62 | 44.57 | 83.96 | 0.80 | 0.0194 | N.A. | 0.52 | 1.47 | 1.63 |
| 63 | 45.58 | 84.29 | 0.81 | 0.0264 | N.A. | 0.71 | 0.78 | 1.41 |
| 64 | 46.68 | 83.54 | 2.01 | 0.0194 | 5.54 | 1.96 | 0.49 | 1.21 |
| 65 | 46.49 | 85.04 | 0.64 | 0.0291 | N.A. | 0.77 | 0.82 | 1.39 |
| 66 | 46.88 | 85.92 | 0.15 | N.A. | N.A. | 0.39 | 0.84 | 1.61 |
| 67 | 47.26 | 86.71 | 10.57 | 0.0169 | 4.41 | 1.47 | 0.63 | 1.30 |
| 68 | 48.15 | 86.56 | 0.57 | 0.0301 | N.A. | 0.59 | 1.10 | 1.54 |
| 69 | 47.86 | 86.29 | 1.41 | 0.0221 | 7.70 | 1.02 | 1.04 | 1.48 |
| 70 | 48.33 | 87.13 | 0.21 | 0.0351 | N.A. | 0.41 | 0.62 | 1.39 |
| 71 | 48.07 | 87.03 | 0.61 | 0.0255 | N.A. | 0.66 | 1.01 | 1.52 |
| 72 | 47.79 | 87.56 | 1.49 | 0.0259 | N.A. | 1.05 | 1.20 | 1.41 |
| 73 | 47.55 | 88.61 | 0.30 | 0.0381 | N.A. | 0.54 | 1.12 | 1.45 |
| 74 | 47.63 | 88.40 | 0.55 | 0.0337 | N.A. | 0.63 | 1.20 | 1.49 |
| 75 | 47.58 | 88.78 | 0.81 | 0.0376 | N.A. | 0.74 | 0.79 | 1.34 |
| 76 | 47.17 | 88.70 | 1.21 | 0.0389 | N.A. | 0.89 | 0.97 | 1.47 |
| 77 | 47.27 | 89.97 | 1.50 | 0.0301 | N.A. | 0.80 | 0.71 | 1.25 |
| 78 | 46.85 | 90.32 | 2.32 | 0.0221 | 5.52 | 1.93 | 0.48 | 1.25 |
| 79 | 43.53 | 89.74 | 3.13 | 0.0221 | 5.52 | 1.65 | 0.46 | 1.20 |
| 80 | 44.10 | 87.49 | 0.54 | 0.0292 | N.A. | 0.49 | 0.74 | 1.66 |
| 81 | 43.60 | 87.51 | 0.72 | 0.0255 | N.A. | 0.45 | 0.80 | 1.65 |
| 82 | 44.09 | 84.80 | 0.23 | N.A. | N.A. | 0.16 | 0.88 | 1.92 |
| 83 | 43.93 | 85.41 | 0.31 | N.A. | N.A. | 0.27 | 0.77 | 1.60 |
| 84 | 43.93 | 86.76 | 1.65 | 0.0129 | 6.66 | 0.95 | 0.79 | 1.52 |





**Table 2.** Description of the three PARAFAC components. The secondary peaks are shown in brackets.

| Component number | Excitation maximal wavelength (nm) | Emission maximal wavelength (nm) | Descriptions | References |
|---|---|---|---|---|
| C1 | <240 (305) | 453 | Terrestrial humic-like substances | Stedmon and Markager, 2005a; Stedmon et al., 2003 |
| C2 | <240 (300) | 393 | Microbial, anthropogenic or terrestrial humic-like substances | Murphy et al., 2011;Zhang et al., 2010 |
| C3 | <240 (270) | 315 | Tyrosine-like fluorophore | Yu et al., 2015 |



**Table 3.** Regional average of relative intensity (in percent) and $F_{max}$ of each fluorescence component.

| Regions | C1 (%) | C2 (%) | C3 (%) | $F_{max}$(C1) (RU) | $F_{max}$(C2) (RU) | $F_{max}$(C3) (RU) |
|---------|--------|--------|--------|--------------------|--------------------|--------------------|
| 1 | 35±4 | 41±1 | 25±5 | 0.13±0.03 | 0.15±0.03 | 0.09±0.01 |
| 2 | 25±11 | 38±8 | 37±16 | 0.11±0.08 | 0.14±0.08 | 0.11±0.02 |
| 3 | 9±5 | 47±13 | 45±16 | 0.03±0.03 | 0.15±0.11 | 0.11±0.01 |
| 4 | 17±10 | 41±6 | 42±10 | 0.08±0.09 | 0.14±0.11 | 0.13±0.10 |
| 5 | 17±8 | 44±7 | 38±8 | 0.05±0.04 | 0.13±0.06 | 0.10±0.02 |
| Total | 20±11 | 42±9 | 38±14 | 0.08±0.07 | 0.14±0.08 | 0.11±0.05 |





**Table 4.** The regional average of fluorescence-derived indices.

| Regions | HIX | BIX | FI |
|---------|-----------|-----------|-----------|
| 1 | 2.21±0.42 | 0.49±0.05 | 1.29±0.05 |
| 2 | 1.42±0.84 | 0.61±0.10 | 1.37±0.15 |
| 3 | 0.62±0.37 | 0.93±0.25 | 1.60±0.15 |
| 4 | 0.91±0.52 | 0.82±0.21 | 1.43±0.12 |
| 5 | 0.94±0.43 | 0.92±0.25 | 1.38±0.09 |
| Total | 1.21±0.78 | 0.76±0.26 | 1.42±0.16 |



**Table 5.** Summary of fluorescence-derived indices from natural water and water extraction of aerosol reported by other studies with average values for some studies are shown in brackets.

| Study area | Sample type | HIX | BIX | FI | References |
|---|---|---|---|---|---|
| Tibetan Plateau | Cryoconite in glaciers | 1.11-1.37 (1.27) | 0.65-0.93 (0.80) | 3.12-3.44 (3.24) | Feng et al., 2016 |
| Yungui Plateau, China | Inland lakes | 0.23-6.00 (1.57) | 0.60-1.54 (0.93) | 1.14-1.80 (1.37) | Zhang et al., 2010 |
| Frasassi Caves, Italy | Cave water | 1.79-3.28 (2.32) | 0.80-1.12 (0.95) | ~1.8 | Birdwell and Engel, 2010 |
| Springs in USA | Spring water | 0.36-1.21 (0.76) | 0.64-1.13 (0.87) | 1.92-2.28 (2.09) | Birdwell and Engel, 2010 |
| Gironde Estuary, France | Estuary | ~10-17 | 0.6-0.8 | 1.14-1.22 | Huguet et al., 2009 |
| North Pacific Ocean | Ocean water | 0.92-1.80 (1.49) | 0.88-1.38 (1.0) | 1.54-1.77 (1.66) | Helms et al., 2013 |
| Tai Mountain, China | Fog water | 3.23-6.79 (4.8) | 0.64-1.02 (0.87) | 1.42-1.83 (1.63) | Birdwell and Valsaraj, 2010 |
| Jianghan Plain, China | Ground water | 2.71-7.49 | 0.88-0.97 | - | Huang et al., 2015 |
| Colorado, USA | Aerosol in alpine sites | 0.72-4.75 (2.42) | 0.54-0.75 (0.65) | 1.18-1.57 (1.4) | Xie et al., 2016 |
| Granada, Spain | Urban aerosol | 2.79-4.89 | - | 1.48-1.61 | Mladenov et al., 2011 |



**Table 6.** Pearson's correlation coefficients (r) of major ions and $F_{max}$ for fluorescence components when excluding data from site 67; the results for the entire dataset are shown in parentheses. Note: [*] denotes $p < 0.001$.

|  | $SO_4^{2-}$ | $NO_3^-$ | nss-$K^+$ |
|---|---|---|---|
| $F_{max}$(C1) | 0.01 (0.14) | -0.10 (-0.04) | 0.23 (0.48) |
| $F_{max}$(C2) | 0.70[*] (0.72) | 0.60[*] (0.57) | 0.63[*] (0.73) |
| $F_{max}$(C3) | 0.44 (0.42) | 0.34 (0.23) | 0.29 (0.68) |



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
