# Peer review of "The optical characteristics and sources of chromophoric dissolved organic matter (CDOM) in seasonal snow of northwestern China"

_The Cryosphere, 2018_

## Referee Comment (RC1) · Anonymous Referee #1 · 30 Jul 2018

The comment was uploaded in the form of a supplement:
https://www.the-cryosphere-discuss.net/tc-2018-122/tc-2018-122-RC1-supplement.zip

---

## Referee Comment (RC2) · Anonymous Referee #2 · 29 Aug 2018

Review of "The optical characteristics and sources of chromophoric dissolved organic matter (CDOM) in seasonal snow of northwestern China" by Zhou et al, submitted to The Cryosphere

General comments

This work describes the results of a field campaign on surface snow chemical properties, lead in northwestern China. More specifically, it investigates the colored dissolved organic matter (CDOM) in seasonal snow, trying to evaluate its different components, their sources, and overall importance for light absorption by snow, which is a very important topic, linked to the climate impact of snow. The subject treated is thus highly

relevant for The Cryosphere and is worth publishing, once the authors take care of the following remarks.

My major issue with this work is related to clarity. Although generally well written, I truly think the authors should make a distinct effort on two aspects: - Have a clearer presentation of the PARAFAC method. Although as stated by the authors, it has become very mainstream in the aquatic chemistry community, and is starting to be used in the aerosol community, it is still a novelty for most readers of The Cryosphere. It would be good, in this case, to have a clear reminder of what the PARAFAC methods gives (what are the Components, the Fmax, . . .). If the authors are somewhat familiar with the PMF method, they might even want to draw a parallel, which might (arguably) help - Maybe draw a clearer separation between actual results, and their interpretation in terms of sources and comparison with previous studies. This could be done by adding a "discussion" section. In particular, this might help clarifying the case on sources. In its current form, the paper discusses sources through the analyses of PARAFAC components, and then though the analyses of backtrajectories and other data such as the ion data. I feel that the case of the authors on sources would much stronger if raw results were presented first (PARAFAC components, clusters, ion ratios, maybe backtrajectories) and then discussed together: this would help synthesis, and avoid losing the reader between two different discussions on the same topics.

Specific comments:

P7 line 14: there were studies on EEM application to aerosols before the recent papers cited here. Please refer to the appropriate literature (probably not exhaustive): - Duarte, R. M. B. ., Pio, C. A. et Duarte, A. C.: Synchronous scan and excitation-emission matrix fluorescence spectroscopy of water-soluble organic compounds in atmospheric aerosols, Journal of Atmospheric Chemistry, 48(2), 157–171, 2004. - Lee, H. J. (Julie), Laskin, A., Laskin, J. et Nizkorodov, S. A.: Excitation–Emission Spectra and Fluorescence Quantum Yields for Fresh and Aged Biogenic Secondary Organic Aerosols, Environ. Sci. Technol., 47(11), 5763‑5770, doi:10.1021/es400644c,

2013. - Mladenov, N., Alados-Arboledas, L., Olmo, F. J., Lyamani, H., Delgado, A., Molina, A. et Reche, I.: Applications of optical spectroscopy and stable isotope analyses to organic aerosol source discrimination in an urban area, Atmospheric Environment, 45(11), 1960‑1969, doi:10.1016/j.atmosenv.2011.01.029, 2011.

P8 line 18: "grouping scheme presented by Pu et al. (2017)": Pu et al actually just refer to "geographical distribution" as a grouping scheme. Reading further in the article, there is a logic, in terms of north or south of a mountain range, on this or that side of strong potential human sources, . . . I suggest this logic should be somewhat detailed here, rather than reporting to a reference where it is actually not clearly presented

P10 line 9-10: "In addition, the emission wavelengths longer than 650 nm were removed to eliminate the uncertainty of measurement". It is not clear what uncertainty is eliminated here. Please be more specific.

P10 line 10-15: as in any statistical factor analysis (PARAFAC, PMF, . . .) the choice of factor number is quite critical, and thus must be very carefully argumented. Here, the choice of 3 components is based on residual error analysis. Yet, although going from 2 to 3 decrases strongly this error, there is still (fig S1) a large error around 270nm which desapears when going from 4 to 5 factors. The authors "confirm" the 3 factor analysis with some splitting method, but they do not reject the 4 or 5 factor analysis with this method. To me, it seems at this point, the choice of 3 factors is largely arbitrary.

P10: I may have missed it, but did the authors mention the number of samples used in the PARAFAC analysis? Is that number sufficient for such a statistical method? There are quality guidelines for PMF studies from filters and offline tracers analysis (Belis et al, 2014), I would expect similar guidelines to exist for PARAFAC, as these methods are mathematically very close (if not equivalent)

P11 line 17-19: the reason put forward by the authors for choosing one fitting method rather than the other seems statistically weak to me. Of course, as any point where the choice would actually matter is anyway rejected in advance, it is of minor importance.

[Figure]

Yet, could the authors be more specific here ?

P11 line 17-25 : confidence on fits ? this translates in uncertainties in the reported slopes and AAE, which might impact interpretations. So it is of some importance !

P12 line 20: nss-K is reportedly calculated after Tao et al 2016 following nss-K = K - 0.159 Mg. Tao et al 2016 actually claim they used Cheng et al, 2000 definition of nss-K, reported as nss-K = K - 0.037 Na, which they claim they took from Hitchcock et al, 1980. I would suggest being more precise on the calculation really made, and its origin. In any case, I also have some reservations on this approach, as it was originally used for coastal sites (North Carolina for Hichcock et al, 1980 ; Hong Kong for Cheng et al, 2000). In more continental areas, with significant input of terrestrial dust, there might be a sizable portion of either $Mg^{2+}$ and/or $Na^+$ coming from dust, which would distort the relation used. An example of such distortion and the way to analyse it is presented in Pio et al, 2007. I suggest to take these points into account.

P13 line 7-8 : how was the choice of 4 clusters decided to be relevant ?

Fig 5: replace "excitaition" with "excitation" ; it is not quite clear to me what "excitation (emission) loading" is. I suspect it is the sum over emission (excitation) of the EEM matrix of each component. Am I right? maybe it should be precised either here in the caption, or somewhere in the text (or both).

P19 line 4-8: It is not clear to a non-PARAFAC aficionado what Fmax is. I feel that it is the fraction of the observed total fluorescence that is explained by a given component, so that it should be somewhat proportional to the concentration of this component in the mixture that the sample is made of. It would be really helpful to clarify this point.

P21 line 9: the picture presented by the authors looks much like lichens to me, more than algae. I also thought that algae only lived in aquatic media (including snow), except when associated with fungi in lichens.

P22 line 18: I feel that figure 9 is not very informative as such

P23 line 2-5: Figure S6 should be considered as a replacement for figure 10: it is more informative, while not being really more clutered. Maybe it could be made even better by reducing slightly the font size of the red equations and drawing dashed red lines, thus using size and line type and color to make stress which correlation is important and which one is rejected.

P25 line 20: see my previous comment on nss-K+. how is the authors discussion here sensitive to the objection raised in that previous comment?

P26: I have a hard time evaluating whether the backtrajectory analysis presented her is relevant. Obviously, fires location from MODIS can be active or not when a back-trajectory passes over it. Here, it seems that any backtrajectory can overpass any fire location, and be taken into account in the analysis, even if the backtrajectory over-passes on day n, and the fire was only active from day n+3 to day n+10. This seems to weaken a lot the analysis presented here.

Belis, C. A., Favez, O., Harrison, R. M., Larsen, B. R., Amato, F., El Haddad, I., Hopke, P. K., Nava, S., Paatero, P., Prévôt, A., Quass, U., et al.: European guide on air pollution source apportionment with receptor models., Publications Office, Luxembourg. Available from: http://dx.publications.europa.eu/10.2788/9307, 2014.

Cheng, Z. ., Lam, K. ., Chan, L. ., Wang, T. et Cheng, K. .: Chemical characteristics of aerosols at coastal station in Hong Kong. I. Seasonal variation of major ions, halogens and mineral dusts between 1995 and 1996, Atmospheric Environment, 34(17), 2771‑2783, doi:10.1016/S1352-2310(99)00343-X, 2000.

Hitchcock, D. R., Spiller, L. L. et Wilson, W. E.: Sulfuric acid aerosols and HCl release in coastal atmospheres: Evidence of rapid formation of sulfuric acid particulates, Atmospheric Environment (1967), 14(2), 165‑182, doi:10.1016/0004-6981(80)90275-9, 1980.

Pio, C. A., Legrand, M., Oliveira, T., Afonso, J., Santos, C., Caseiro, A., Fialho, P.,

Barata, F., Puxbaum, H., Sanchez-Ochoa, A., Kasper-Giebl, A., et al.: Climatology of aerosol composition (organic versus inorganic) at nonurban sites on a west-east transect across Europe, J. Geophys. Res., 112(D23), doi:10.1029/2006JD008038, 2007.

---

## Author Comment (AC1) · 25 Oct 2018

Response to reviewer#1

We are very grateful for the reviewer's critical comments, which have helped us improve the paper quality substantially. We have addressed all of the comments carefully as detailed below in our point-by-point responses. Our responses start with "R:".

**General comments**

This paper aims at studying the optical characteristics and sources of CDOM in the seasonal snow. The paper is generally well written. Please find below my general comments.

R: Thanks for the reviewer's comments, we have addressed all of the comments carefully as detailed below.

1. A large proportion of the introduction is devoted to present various usages of spectrometry indices. Rather, authors should use this space to better present the problems they are trying to address.

R: We have totally rewritten the introduction. The discussions about the CDOM in aquatic environments and the usages of spectrometry indices have been weakened, and replaced by the scientific progress on the characteristics of DOM and CDOM in the cryosphere.

2. In the methods section, authors said they frozen water samples before optic measurements. Freezing DOM samples are problematic because of sedimentation/precipitation processes that are further causing scattering (Thieme et al. (2016); Fellman, D'Amore, and Hood (2008)). Authors need to carefully address this issue. To cite Fellman, D'Amore, and Hood (2008):

   We further show that when surface water samples were frozen, there was a decrease in the specific ultraviolet absorbance (SUVA) of DOC that is particularly evident

with high concentrations of DOC.

R: We have added the discussion of the uncertainties due to the freeze-thaw process into the method section (lines 5-22, page 8). We agreed with the reviewer that the freeze-thaw process may lead to biases of the optical properties for the DOM samples. According to the previous studies, Fellman et al. (2008) reported that there was a decrease of specific ultraviolet absorbance (SUVA) for stream water DOM after frozen, with a median of approximately 8%. A study of peatland DOC found that the change of light absorption at 254 nm after freeze and thaw was less than 5% in median (Peacock et al., 2015). Thieme et al. (2016) assessed the changes of fluorescence properties for several types of DOM samples. The results showed the decreased relative percentages of terrestrial humic-like fluorophores (-3% on average) and HIX (-2% on average), and the increased percentage of fluvic-like fluorophore (+6% on average). However, various types of DOM in previous studies were shown that their optical properties (light absorption and fluorescence) were not affected significantly by frozen effect, such as ocean water, pore water, spring and cave water (Birdwell and Engel, 2010; Del Castillo and Coble, 2000; Otero et al., 2007; Yamashita et al., 2010). As discussed above, the freeze-thaw process may influence the relative contributions of PARAFAC components slightly, and the effects on $a_{280}$ and the fluorescence indices can be neglected. It seems that the impact of freezing to optical properties of DOM samples varies largely with the sample types, preservation methods, DOC concentrations and optical parameters. There is limited study focuses on the preservation effects on snow DOM, which is frozen in the nature. Therefore, it is urgent to fill in this gap to minimize the artifacts of freezing in future studies.

3. Many figures in the manuscript are used to present the relationships among the calculated optical indices. These do not contribute to increasing our knowledge about DOM in the snowpack. Actually, there are many studies that compared optical indices. Hence, these figures are not interesting in the context of the current study. Authors should carefully review the objectives of the paper and use appropriate figures.

R: We have replotted nearly all of the figures except Fig. 8 and Fig. 13. We have also rewritten the introduction, and restructured the results and discussion section. The discussions about the sources of CDOM have been moved to Sec. 3.3 in the revised manuscript. We hope that these modifications can improve the quality of our manuscript significantly.

4. In relation with my previous comment, I found the ratio between the length of the paper and new knowledge to be rather high. I believe that the results/discussion section could be shortened by at least 50%.

R: We have shortened the results and discussion section.

**Specific comments**

**Introduction**

5. Page 4, line 4: Helms et al. (2008) did not show that. It was already known in the 1960's. This is the same for the next citation.

R: We have removed the citation and rewritten lines 3-6 in page 3, as follows: "Chromophoric dissolved organic matter (CDOM), widely known as the light-absorbing constituent of DOM, can absorb light from ultraviolet to visible (UV-vis) wavelengths (Bricaud et al., 1981)."

6. Page 5, line 16: Do not start a paragraph with however.

R: We have rewritten the introduction and modified the grammar mistake.

7. Page 6, lines 10-14: Please review the sentences (2 times however). I do not understand the sentence at line 10. I thought you were talking about CDOM in the snow, not in the atmosphere or in water bodies.

R: (1) We have rewritten the introduction, and the improper expression have been removed.

(2) Sorry for the misleading, we indicated that the CDOM had been widely studied

in aerosol and water bodies, but rarely investigated in seasonal snow. The corresponding description has been changed in lines 10-12, page 4, as follows:

"However, these studies neglected CDOM, which is rarely studied in snow but has been proved as an effective light absorber whether in the atmosphere (i.e., brown carbon, BrC) (Hecobian et al., 2010) or water bodies (Bricaud et al., 1981)."

8. Page 6, line 20: Why 280 nm? This is rather unusual in the literature. Most people use either 275 nm or 254 nm.

R: We agreed with the reviewer, and the sentence has been modified in line 20, page 5 to line 1, page 6, as follows:

"The absorption coefficient at a certain wavelength within the UV band, for instance, 254 nm, 280 nm or 350 nm (Spencer et al., 2012; Zhang et al., 2010, 2011), usually serve as an indicator of CDOM abundance.".

9. Page 6, lines 17: This paragraph on the use UV-vis spectroscopy could be shortened.

R: The description of the UV-vis and fluorescence spectroscopy has been combined together and shortened greatly. Please see line 19, page 5 to line 17, page 6 in the revised manuscript.

10. Page 7, lines 9-14: These advantages are also valid for absorption measurements.

R: The advantages of fluorescence and absorption measurements has been rewritten in lines 19-20, page 5, as follows:

"UV-vis absorption and fluorescence spectroscopies are both rapid and effective methods of characterizing the optical properties and sources of CDOM."

11. Page 7, lines 15-16: Authors should be a clear distinction between CDOM and FDOM (EEMs). FDOM is a sub-fraction of the CDOM.

R: The "CDOM" has been replaced by "fluorescent DOM (FDOM)" in line 8, page 6.

**Methods and materials**

12. Page 9, lines 10-13: Freezing DOM samples is problematic. See my general comments. Also, at line 15, it is said that samples were analyzed within 24h. It is not clear how samples were processed.

R: (1) Please see our response to comment 2.

(2) The procedure of snow samples has been described in more detail in lines 2-7, page 9, as follows:

"The snow samples were firstly melted under the room temperature. Then, the snow water samples were filtrated using 0.22 μm PTFE syringe filters (Jinteng, Tianjin, China), and stored in prebaked glass vials (450 °C for 4 h) at 4 °C in a freezer. All the samples were measured for UV-Vis and fluorescence spectroscopies within 24 hours after filtration. The ultrapure water (18.2 MΩ·cm) filtrated by the PTFE syringe filters exhibited no clear fluorescence signal."

13. Page 9, line 19:  measured

R: Changed as suggested in line 8, page 9.

14. Page 9, line 21:8 What are the 8 pixels? I never heard about that term.

R: In the software of Aqualog spectrofluorometer, the increment of emission wavelength was measured in pixels with a conversion to nm, as shown in the black box of the following picture. To minimize the readout noise of the CCD detector, we use the maximum increment (8 pixels, 4.65 nm) in the experiment.

[Figure]

15. Page 10: Equations are not written correctly. For example, $I_{370}^{450}$ is the integrated fluorescence between these two wavelengths. Please use the appropriate notation.

R: We have changed the notations in the Eq. (1-3) as follows:

FI = I (Ex = 370, Em = 450) / I (Ex = 370, Em = 499);

BIX = I (Ex = 310, Em = 379) / I (Ex = 310, Em = 430);

HIX = I (Ex = 255, Em = 434-480) / I (Ex = 255, Em = 300-345);

16. Page 11, lines 16-21: How the exponential fit was performed? Was a background coefficient K used? If so it is problematic to fit a non-exponential function of such narrow spectral range with a background coefficient.

R: The equation used in this study has been presented as Eq. (5): $a(\lambda) = a(\lambda_r)e^{-S(\lambda - \lambda_r)}$, given by Twardowski et al. (2004). The background coefficient K was not used to perform the exponential fit. This equation and its corresponding description have been added in line 21, page 11 to line 2, page 12.

17. Page 12, equation 5: More details should be given about this metric since it is not widely used by the community.

R: More details have been added in lines 6-9, page 12, as follows:

"The absorption Ångström exponent (AAE) is used to describe the wavelength dependence of light absorption for aerosol (Bond, 2001), which was also applied to characterize the ILAPs and CDOM in snow and ice (Doherty et al., 2010; Niu et al., 2018; Wang et al., 2013; Yan et al., 2016)."

18. Page 13, lines 3-8: It is not clear how the clustering was performed. What are the multiple correlation coefficients?

R: Sorry for the misleading. The "multiple correlation coefficients" in the text does not represent a term, we indicated the various cophenetic correlation coefficients for different clustering methods (e.g., weighted average method, centroid method, and so on). The cophenetic correlation coefficient is a criterion of the efficiency of clustering methods (Saracli et al., 2013). Higher cophenetic correlation coefficient indicates that the clustering method is better. Sec. 2.5 has been rewritten in lines 1-8, page 14.

19. Page 13, lines 9-17: This is the first time authors talk about fires. How is it related to the current study? This is an example where authors should better use the introduction to present the problem and what they did to address it.

R: Sec. 2.6 has been rewritten, and the details about air mass backward trajectories and fire location map have been added in line 10, page 14 to line 2, page 15.

**Results and discussion**

20. Figure 3: Why results from Qinghai region are not presented?

R: All the samples sites in Xinjiang and Qinghai are shown in Fig. 3. To avoid such ambiguity, we have updated Fig. 1 and Fig. 3. We believe that the revised figures could be much friendlier to the readers who are not familiar with the geography of China.

[Figure]

**Figure 1.** (a) The location of study area and sample site distribution across northwestern China. The site numbers and regional groupings are shown in panel (b) for Xinjiang and (c) for Qinghai. Sample areas are divided into five regions indicated by different symbol shapes, and the land cover types of sample sites are represented in different colors, as shown in the legend in panel (a). The "D" indicates that the sample was collected from a snow drift, and the "F" indicates that the surface sample was fresh snow. The elevation is shown in the contour plot.

[Figure]

**Figure 3.** $a_{280}$ and $S_{275-295}$ for sites in (a, c) Xinjiang and (b, d) Qinghai, respectively. The five regions are indicated by different symbols (same as Fig. 1).

21. Page 14, lines 1-2: In Fig. 3, a280 varies between 0 and 4.5, not between 0.15 and 10.57 as said in the text.

R: Sorry for the misleading. The symbols in black color shown in Fig. 3 represents $a_{280}$ higher than 3.5 m$^{-1}$. To make the figure more clearly, the color bar has been updated in the revised manuscript.

22. Page 14, lines 13-15: Why comparing S measured in snow and S measured in oceans? This sentence is detached from the rest of the text.

R: The values of $S_{275-295}$ were rarely reported in the cryosphere in previous studies. Hence, we compared our results to the values of various types of aquatic environments

summarized by Hansen et al. (2016). There is a large difference of $S_{275-295}$ between oceanic and terrestrial systems (0.020-0.030 $nm^{-1}$ and 0.012-0.023 $nm^{-1}$, respectively) due to the different CDOM sources. We noted that the $S_{275-295}$ in snow showed a broad range of 0.0129-0.0389 $nm^{-1}$, which covered the value ranges of different aquatic environments, may indicating complex sources. We have rewritten these sentences in line 17, page 15 to line 3, page 16, as follows:

"$S_{275-295}$ is never reported in the terrestrial snow and ice samples before, but is widely measured in the aquatic environments. For example, Hansen et al. (2016) summarized the $S_{275-295}$ for oceanic and terrestrial systems, the values range of 0.020-0.030 $nm^{-1}$ for ocean, 0.010-0.020 $nm^{-1}$ for coastal water, and 0.012-0.023 $nm^{-1}$ for terrestrial systems. The $S_{275-295}$ in this study covered the typical values in different types of natural water bodies, indicating complex compositions and sources of CDOM in seasonal snow across northwestern China."

23. Figure 4: There is a relation (which is already known in the literature) between S275-350 and a280. What does it mean in the context of this study? As I said, this relation is already known, so I am not sure that this figure is needed.

R: We agreed with the reviewer. Fig. 4 and the corresponding description have been removed.

24. Page 14, line 22: What is HULIS?

R: "HULIS" is the abbreviation of "humic-like substances". We have updated this information in the introduction in line 20, page 4 to line 1, page 5, as follows:

"while humic-like substances (HULIS), which is a type of macromolecular organic substances defined for aerosol with certain similar chemical properties to terrestrial and aquatic humic and fulvic substances (Graber and Rudich, 2006), and unknown chromophores each accounted for approximately half of the total absorption."

25. Page 15, lines 2-6: Why AAE values not presented in a map like for S and a?

R: The AAE values were calculated from 240-550 nm. Because the light absorption

within the visible wavelengths of some samples were below the detection limit of the spectrometer, approximately half of the samples were available for the AAE calculation. This has been mentioned in lines 15-17, page 12. Due to the missing values appeared, the AAEs were not shown as a figure in this study but summarized in Table 1, which could be useful for further studies.

26. Pages 15-17: These results are site specific and cannot be generalized. The Editor should check if this is in line with the scope of the journal. Since authors are interested in presenting differences among regions, I suggest using boxplots instead of Fig. 3 and Fig. 7. Then, ANOVA or t-test could be used.

R: We have replotted Fig. 3 and Fig. 7. The results in pages 16-17 were also revised. We tried to assess the optical properties and sources of snow CDOM across northwestern China, not only the differences among regions. The sample sites were grouped based on the geographical distributions, because we suggested that the geographical locations combined with the local land cover and topography might be the major mechanisms for the variations of sources and optical properties of CDOM in snow. Our previous study has proved that the sources of insoluble light-absorbing particles show regional variations in the same field campaign (Pu et al., 2017). Additionally, in the following discussion (Sec. 3.3), we really found variations of sources for snow CDOM among some regions, for instance, regions 1, 3 and 4; meanwhile, some regions also showed similar characteristics, like regions 1 and 2, regions 4 and 5. As suggested by the reviewer, ANOVA have been used to assess the differences among regions in Sec. 3.3.

27. Section 3.3: These results are not related to paper objectives that aim to study DOM in the snow.

R: We agreed with the reviewer. Sec. 3.3 has been removed in the revised manuscript.

28. Table S1 same results as in Fig. S5.

R: We have visualized Table S1 as pie charts and added them into Fig. S4 in the revised

Supplement.

[Figure]

**Figure S4.** (a-d) The results of cluster analysis, and (e-h) the average %C1-%C3 in each cluster (pies).

29. Fig. 5: This should be in the appendix.

R: We should admit that the fingerprints of EEM components decomposed by PARAFAC method were widely used to discuss the CDOM in aquatic environments. However, the potential readers of this manuscript are likely to be the scientists who are expert in the cryosphere. The EEMs combined with the PARAFAC analysis is rarely used in this research field, and Fig. 5 (Fig. 4 in the revised manuscript) can give a visualization of the fluorescent components appeared in the snow of northwestern China. Therefore, we have kept the panels a-c in the main text and removed the panels d-f.

30. Section 3.2.1: Three pages are dedicated to present PARAFAC components. Once

again, what kind of information this brings in the context of the paper?

R: We have shortened Sec 3.2.1, and the correlation analysis among PARAFAC components has been moved to Sec. 3.3 in the revised manuscript. The EEM combined with the PARAFAC analysis is the key analytical tool in this study. Since these components do not appear frequently in the studies of snow and ice, we suggested that the thorough discussion of the present knowledge of PARAFAC components is needed. Besides, the correlation analysis is a useful method to identify the potential sources of the PARAFAC components (e.g., Murphy et al., 2008; Zhang et al., 2011), which is also very important in this study.

31. Fig. 7: This figure is very difficult to interpret. It is rarely a good idea to present pie chart because human eyes are very bad at judging angles. I suggest a figure like this:

[Figure]

With such figure, regional variations will be better visualized.

R: Fig. 7 has been replotted as Fig. 5 in the revised manuscript, and the information from Table 3 has been added into Fig. 5 as pie charts.

[Figure]

**Figure 5.** Variations of the fluorescent components among regions. The boxplots show the intensities of components. The boxes denote the 25th and 75th quantiles, and the horizontal lines represent the 50th quantiles (medians), the averages are shown as dots; the whiskers denote the maximum and minimum data within 1.5 times of interquartile rang, and the datapoints out of this range are marked as "+". The pie charts show the average relative contributions of three components in each region. C1, C2, and C3 are represented in red, yellow, and blue, respectively, both for the boxplots and pie charts. The percentages on the left of the panel are the averages of %C1-%C3 for the whole dataset.

32. Figure 8: The clustering should be done using all optical indices (S, AAE, a280).

R: Due to missing values of S and AAE, cluster analysis is not available for such parameters. In this study, the cluster analysis is used to assess the compositions or sources variations among samples, however, $a_{280}$ does not contain such information and was not involved into the cluster analysis. Furthermore, in previous studies, the input dataset of cluster analysis was usually derived from the same types of measurement, such as EEM-PARAFAC (Dubnick et al., 2010; Maie et al., 2012; Zhao et al., 2016), Fourier transform infrared (FT-IR) spectroscopy (Yang et al., 2015) or high-resolution mass spectrometry (HR-MS) (Chen et al., 2016). Therefore, we used the relative contributions of three fluorescent components in the cluster analysis.

33. Figure 9: Any reasons to present sites in that specific order? This can be confusing

if there is no link among regions. What are the a and b letters under stations 51 and 52?

R: (1) These sites were grouped by regions and arranged in the order of regions 1 to 5 in Fig. 9. We have updated Fig. 9 to the boxplot as Fig. 7 in the revised manuscript using the data in Table 4 (Table 4 has been removed). The values in each site can be found in Table 1. In this way, the variations among regions could be clear.

(2) To assess the variations of CDOM properties in a same snowpack, we collected two snow profiles at site 51 marked as sites 51a and 51b. The results showed that the properties of CDOM at sites 51a and 51b were quite similar.

[Figure]

**Figure 7.** HIX (shown in red), BIX (shown in blue) and FI (shown in green) of surface snow samples among regions. The meaning of each part of box is same as that in Fig. 5.

34. Figure 10: See my other comments about showing how optical indices compare and the aim of the study.

R: Sec.3.3 has been removed as mentioned in our response to comment 27, correspondingly, Fig.10 has also been deleted in the main text. Fig.10 (a-c) has been moved to the Supplement as Fig. S5.

35. Tables 3, 4 and 5: This data could be presented using boxplots are better than tables for visualization. Raw data should be given in the appendix.

R: As shown in our responses to comments 31 and 33, the data in Table 3 has been added into Fig. 5 in the revised manuscript, and Table 4 has been plotted as Fig. 7 in the revised manuscript based on the reviewer's suggestions. For Table 5, we have also

drawn a boxplot for visualizing the data as the following figure shown, however, we noted that Table 5 could also provide useful information for the comparison among studies. Hence, we suggested that Table 5 could be retained in the revised manuscript if the reviewer also agreed.

[Figure]

References:
A: Cryoconite in glaciers on the Tibetan plateau. Feng et al., 2016.
B: Inland lakes on the Yungui Plateau, China. Zhang et al., 2010.
C: Frasassi Cave water, Italy. Birdwell and Engel, 2010.
D: Springs in USA. Birdwell and Engel, 2010.
E: Gironde Estuary, France. Huguet et al., 2009.
F: North Pacific Ocean. Helms et al., 2013.
G: Fog water collected on the Mt. Tai, China. Birdwell and Valsaraj, 2010.
H: Ground water in Jianghan Plain, China. Huang et al., 2015.
I: Aerosol in alpine sites, Colorado, USA. Xie et al., 2016.
J: Urban aerosol, Granada, Spain. Mladenov et al., 2011.

**References**

Birdwell, J. E., and Engel, A. S.: Characterization of dissolved organic matter in cave and spring waters using UV-Vis absorbance and fluorescence spectroscopy, Org. Geochem., 41, 270-280, 2010.

Chen, Q. C., Miyazaki, Y., Kawamura, K., Matsumoto, K., Coburn, S., Volkamer, R., Iwamoto, Y., Kagami, S., Deng, Y. G., Ogawa, S., Ramasamy, S., Kato, S., Ida, A., Kajii, Y., and Mochida, M.: Characterization of Chromophoric Water-Soluble Organic Matter in Urban, Forest, and Marine Aerosols by HR-ToF-AMS Analysis and Excitation Emission Matrix Spectroscopy, Environ. Sci. Technol., 50, 10351-10360, 2016.

Del Castillo, C. E., and Coble, P. G.: Seasonal variability of the colored dissolved organic matter during the 1994-95 NE and SW Monsoons in the Arabian Sea, Deep-Sea Res. Pt. II, 47, 1563-1579, 2000.

Dubnick, A., Barker, J., Sharp, M., Wadham, J., Lis, G., Telling, J., Fitzsimons, S., and Jackson, M.: Characterization of dissolved organic matter (DOM) from glacial environments using total fluorescence spectroscopy and parallel factor analysis, Ann. Glaciol., 51, 111-122, 2010.

Fellman, J. B., D'Amore, D. V., and Hood, E.: An evaluation of freezing as a preservation technique for analyzing dissolved organic C, N and P in surface water samples, Sci. Total Environ., 392, 305-312, 2008.

Hansen, A. M., Kraus, T. E. C., Pellerin, B. A., Fleck, J. A., Downing, B. D., and Bergamaschi, B. A.: Optical properties of dissolved organic matter (DOM): Effects of biological and photolytic degradation, Limnol. Oceanogr., 61, 1015-1032, 2016.

Maie, N., Yamashita, Y., Cory, R. M., Boyer, J. N., and Jaffe, R.: Application of excitation emission matrix fluorescence monitoring in the assessment of spatial

and seasonal drivers of dissolved organic matter composition: Sources and physical disturbance controls, Appl. Geochem., 27, 917-929, 2012.

Murphy, K. R., Stedmon, C. A., Waite, T. D., and Ruiz, G. M.: Distinguishing between terrestrial and autochthonous organic matter sources in marine environments using fluorescence spectroscopy, Mar. Chem., 108, 40-58, 2008.

Otero, M., Mendonca, A., Valega, M., Santos, E. B. H., Pereira, E., Esteves, V. I., and Duarte, A.: Fluorescence and DOC contents of estuarine pore waters from colonized and non-colonized sediments: Effects of sampling preservation, Chemosphere, 67, 211-220, 2007.

Peacock, M., Freeman, C., Gauci, V., Lebron, I., and Evans, C. D.: Investigations of freezing and cold storage for the analysis of peatland dissolved organic carbon (DOC) and absorbance properties, Environ. Sci-Proc. Imp., 17, 1290-1301, 2015.

Saracli, S., Dogan, N., and Dogan, I.: Comparison of hierarchical cluster analysis methods by cophenetic correlation, J. Inequal. Appl., 203, 10.1186/1029-242X-2013-203, 2013.

Thieme, L., Graeber, D., Kaupenjohann, M., and Siemens, J.: Fast-freezing with liquid nitrogen preserves bulk dissolved organic matter concentrations, but not its composition, Biogeosciences, 13, 4697-4705, 2016.

Twardowski, M. S., Boss, E., Sullivan, J. M., and Donaghay, P. L.: Modeling the spectral shape of absorption by chromophoric dissolved organic matter, Mar. Chem., 89, 69-88, 2004.

Yamashita, Y., Cory, R. M., Nishioka, J., Kuma, K., Tanoue, E., and Jaffe, R.: Fluorescence characteristics of dissolved organic matter in the deep waters of the Okhotsk Sea and the northwestern North Pacific Ocean, Deep-Sea Res. Pt. II, 57, 1478-1485, 2010.

Yang, L., Han, D. H., Lee, B. M., and Hur, J.: Characterizing treated wastewaters of different industries using clustered fluorescence EEM-PARAFAC and FT-IR spectroscopy: Implications for downstream impact and source identification, Chemosphere, 127, 222-228, 2015.

Zhang, Y. L., Yin, Y., Feng, L. Q., Zhu, G. W., Shi, Z. Q., Liu, X. H., and Zhang, Y. Z.: Characterizing chromophoric dissolved organic matter in Lake Tianmuhu and its catchment basin using excitation-emission matrix fluorescence and parallel factor analysis, Water Res., 45, 5110-5122, 2011.

Zhao, Y., Song, K., Li, S., Ma, J., and Wen, Z.: Characterization of CDOM from urban waters in Northern-Northeastern China using excitation-emission matrix fluorescence and parallel factor analysis, Environ. Sci. Pollut. R., 23, 15381-15394, 2016.

---

## Author Comment (AC2) · 25 Oct 2018

Response to reviewer#2

We are very grateful for the reviewer's insightful comments, which are helpful and valuable for greatly improving our manuscript. We have addressed all of the comments carefully as detailed below in our point-by-point responses. Our responses start with "R:".

**General comments:**

This work describes the results of a field campaign on surface snow chemical properties, lead in northwestern China. More specifically, it investigates the colored dissolved organic matter (CDOM) in seasonal snow, trying to evaluate its different components, their sources, and overall importance for light absorption by snow, which is a very important topic, linked to the climate impact of snow. The subject treated is thus highly relevant for The Cryosphere and is worth publishing, once the authors take care of the following remarks.

R: Thanks very much for the reviewer's comments. We have carefully responded the following remarks.

My major issue with this work is related to clarity. Although generally well written, I truly think the authors should make a distinct effort on two aspects:

1. Have a clearer presentation of the PARAFAC method. Although as stated by the authors, it has become very mainstream in the aquatic chemistry community, and is starting to be used in the aerosol community, it is still a novelty for most readers of The Cryosphere. It would be good, in this case, to have a clear reminder of what the PARAFAC methods gives (what are the Components, the Fmax, …). If the authors are somewhat familiar with the PMF method, they might even want to draw a parallel, which might (arguably) help.

R: Thanks for the reviewer's insightful comments. We have added a brief introduction of PARAFAC method in lines 19-22, page 9, and also added the interpretations of the theory and several terms of PARAFAC method in the Supplement, such as the components, $F_{max}$ and loadings. We hope that it can be helpful for the scientists who

are not familiar with this method.

2. Maybe draw a clearer separation between actual results, and their interpretation in terms of sources and comparison with previous studies. This could be done by adding a "discussion" section. In particular, this might help clarifying the case on sources. In its current form, the paper discusses sources through the analyses of PARAFAC components, and then though the analyses of back trajectories and other data such as the ion data. I feel that the case of the authors on sources would much stronger if raw results were presented first (PARAFAC components, clusters, ion ratios, maybe back trajectories) and then discussed together: this would help synthesis, and avoid losing the reader between two different discussions on the same topics.

R: We have reconstructed the results and discussion section. The discussion of CDOM sources have moved to Sec. 3.3.1 and Sec. 3.3.2. Considering the consistency of this paper, the discussion about the optical characteristics (Sec. 3.1, Sec. 3.2, and Sec. 3.4) were still after or together with the results.

**Specific comments:**

3. P7 line 14: there were studies on EEM application to aerosols before the recent papers cited here. Please refer to the appropriate literature (probably not exhaustive):

- Duarte, R. M. B., Pio, C. A. et Duarte, A. C.: Synchronous scan and excitation emission matrix fluorescence spectroscopy of water-soluble organic compounds in atmospheric aerosols, Journal of Atmospheric Chemistry, 48(2), 157–171, 2004.

- Lee, H. J. (Julie), Laskin, A., Laskin, J. et Nizkorodov, S. A.: Excitation–Emission Spectra and Fluorescence Quantum Yields for Fresh and Aged Biogenic Secondary Organic Aerosols, Environ. Sci. Technol., 47(11), 5763-5770, doi:10.1021/es400644c, 2013.

- Mladenov, N., Alados-Arboledas, L., Olmo, F. J., Lyamani, H., Delgado, A., Molina, A. et Reche, I.: Applications of optical spectroscopy and stable isotope analyses to organic aerosol source discrimination in an urban area, Atmospheric Environment,

45(11), 1960-1969, doi:10.1016/j.atmosenv.2011.01.029, 2011.

R: All of the above literatures have been cited in lines 10-11, page 6.

4. P8 line 18: "grouping scheme presented by Pu et al. (2017)": Pu et al actually just refer to "geographical distribution" as a grouping scheme. Reading further in the article, there is a logic, in terms of north or south of a mountain range, on this or that side of strong potential human sources, … I suggest this logic should be somewhat detailed here, rather than reporting to a reference where it is actually not clearly presented.

R: We agreed with the reviewer. Reasons for such grouping scheme have been presented in more detail in lines 5-12, page 7. As the reviewer said, "north or south of a mountain range" or "this or that side of strong potential human sources", actually, the logic for such grouping scheme is still the geographical distribution, we noted that the different geographical locations combined with the land cover and topography can be the major mechanisms in leading to the variously optical properties and sources for CDOM. In addition, our previous study found clearly regional variations of insoluble light-absorbing impurities in the same field campaign (Pu et al., 2017).

5. P10 line 9-10: "In addition, the emission wavelengths longer than 650 nm were removed to eliminate the uncertainty of measurement". It is not clear what uncertainty is eliminated here. Please be more specific.

R: Because the emission signals were mainly within 250-650 nm, those at longer wavelengths were week and more likely to be noises, which might influence the performance of PARAFAC model. The corresponding description has been added in lines 3-6, page 10.

6. P10 line 10-15: as in any statistical factor analysis (PARAFAC, PMF, …) the choice of factor number is quite critical, and thus must be very carefully argued. Here, the choice of 3 components is based on residual error analysis. Yet, although going from 2 to 3 decreases strongly this error, there is still (fig S1) a large error around 270 nm which disappears when going from 4 to 5 factors. The authors "confirm" the 3 factor

analysis with some splitting method, but they do not reject the 4 or 5 factor analysis with this method. To me, it seems at this point, the choice of 3 factors is largely arbitrary.

R: We agreed with the reviewer that the model validation is the most important step in PARAFAC method. The split-half analysis is the most powerful way to confirm the factor numbers of the model (Murphy et al., 2013). The data set is firstly divided into two random, typically equal sized groups and conducting a PARAFAC model on both halves independently. If the correct number of components is chosen, the loadings from both the models will be the same (Stedmon et al., 2003). Many studies used the split-half analysis alone to validate the model (e.g., Yamashita et al., 2008; Zhang et al., 2010; Zhao et al., 2016). Although there were both significant decreases of residual error when component number increased from 2 to 3 and 4 to 5; when subsequently conducting split-half method for 2- to 7-component model, only the 2- and 3-component models passed the analysis. Therefore, the 3-component model was confirmed and the 5-component model was rejected. The corresponding description has been changed in lines 6-11, page 10.

7. P10: I may have missed it, but did the authors mention the number of samples used in the PARAFAC analysis? Is that number sufficient for such a statistical method? There are quality guidelines for PMF studies from filters and offline tracers analysis (Belis et al, 2014), I would expect similar guidelines to exist for PARAFAC, as these methods are mathematically very close (if not equivalent).

R: Yes, there is a recommended sample number for PARAFAC analysis. As shown in Stedmon and Bro (2008), at least 20 samples are required. Of course, if more data is used, easier for getting a robust model. In this study, 78 EEMs were measured, and 76 of them were involved into the PARAFAC model (removed 2 contaminated samples), which is sufficient for this method. We have added the number of measured samples (n = 78) in line 8, page 9 and line 6, page 11.

8. P11 line 17-19: the reason put forward by the authors for choosing one fitting method rather than the other seems statistically weak to me. Of course, as any point where the

choice would actually matter is anyway rejected in advance, it is of minor importance. Yet, could the authors be more specific here?

R: Actually, the variation of results for these two fitting methods was approximately 3% on average. This was consistent with Helms et al. (2008), who first introduced $S_{275-295}$ into the CDOM research field. Due to fits were conducted in a narrow wavelength band (275-295 nm), the data points were much fewer than the whole band. It can lead to a better performance of linear fit than the exponential fit, which was proved by the higher $R^2$ of linear fit. Therefore, we chose the linear fit here. The description has been added in lines 2-3, page 12.

9. P11 line 17-25: confidence on fits? this translates in uncertainties in the reported slopes and AAE, which might impact interpretations. So it is of some importance!

R: $R^2$ of all the fits ($S_{275-295}$ and AAE) were higher than 0.9 and most of them were higher than 0.95. Hence, we noted that these results are credible and accurate. We have added the corresponding description in lines 13-14, page 12.

10. P12 line 20: nss-K is reportedly calculated after Tao et al 2016 following nss-K = K - 0.159 Mg. Tao et al 2016 actually claim they used Cheng et al, 2000 definition of nss-K, reported as nss-K = K - 0.037 Na, which they claim they took from Hitchcock et al, 1980. I would suggest being more precise on the calculation really made, and its origin. In any case, I also have some reservations on this approach, as it was originally used for coastal sites (North Carolina for Hichcock et al., 1980; Hong Kong for Cheng et al., 2000). In more continental areas, with significant input of terrestrial dust, there might be a sizable portion of either Mg2+ and/or Na+ coming from dust, which would distort the relation used. An example of such distortion and the way to analyse it is presented in Pio et al, 2007. I suggest to take these points into account.

R: We agreed with the reviewer and corrected the $K^+$ to nss-ndust-$K^+$, the details can be seen in the main text in lines 5-20, page 13.

11. P13 line 7-8: how was the choice of 4 clusters decided to be relevant?

R: The determination of cluster number is also an important and difficult issue. Because only three parameters were used in the analysis, solutions with too many clusters can lead to difficulty of results interpretation. In this study, 3- to 5-cluster solutions were taken into consideration. The following figure shows the relative contributions of three fluorescent components in each cluster for 3- to 5-cluster solutions. The clusters B and C of 4-cluster solution (panel b) were decomposed from cluster B of 3-cluster solution (panel a). The relative intensities of C2 and C3 in clusters B and C (4-cluster solution) were significantly different (ANOVA, p<0.001), therefore, the 4-cluster solution is appropriate and better than the 3-cluster solution. As for 5-cluster solution (panel c), only sample no. 62 was isolated as the cluster E. The cluster contained very few data should be avoided for determination of cluster numbers. Hence, the 5-cluster solution was rejected., and 4-cluster solution was adopted here.

[Figure]

12. Fig 5: replace "excitaition" with "excitation"; it is not quite clear to me what "excitation (emission) loading" is. I suspect it is the sum over emission (excitation) of the EEM matrix of each component. Am I right? maybe it should be precise either here in the caption, or somewhere in the text (or both).

R: (1) Sorry for the negligence, the wrong word in Fig. 5 has been replaced (Fig. 4 in the revised version).

(2) To explain the term "loading", we should go back to the theory of this method. PARAFAC decomposes a three-way dataset into a set of trilinear terms and a residual array:

$$x_{ijk} = \sum_{f=1}^{F} a_{if} b_{jf} c_{kf} + \varepsilon_{ijk},$$
$$i = 1, ..., I; \ j = 1, ..., J; \ k = 1, ..., K. \tag{R1}$$

In Eq. (R1), $x$ is the original data set, $i$ is the sample number, $j$ and $k$ are the excitation and emission wavelength numbers, respectively; $f$ is the number of fluorescent components, $\varepsilon$ is the residual containing noise and other unmodeled variation. Parameters $a$, $b$ and $c$ represent the concentration, emission spectra and excitation spectra of each fluorophore, respectively. Actually, "loadings" are the parameters $b$ and $c$ calculated from the model, these two parameters showed the basic information of each component. Details can be seen in Stedmon and Bro (2008). Because the term "loading" cannot be explained easily, we have given a brief description in the Supplement (Sec. S1.1), and also deleted Fig. 5d-f since Fig. S2 shows the same information.

13. P19 line 4-8: It is not clear to a non-PARAFAC aficionado what Fmax is. I feel that it is the fraction of the observed total fluorescence that is explained by a given component, so that it should be somewhat proportional to the concentration of this component in the mixture that the sample is made of. It would be really helpful to clarify this point.

R: $F_{max}$ represents the max fluorescence intensity of each component, which is calculated by parameters $a$, $b$ and $c$ in Eq. (R1) and with the same unit of the original EEMs. $F_{max}$ is truly proportional to the concentration of corresponding composition.

However, the intensity of fluorescence does not only dependent on the concentration, as well as the molar absorptivity and fluorescence quantum efficiency. In other words, if species A has higher $F_{max}$ than species B, we cannot conclude that A is more abundant than B. If certain PARAFAC component can be identified as any known chemical species, then the quantification can be performed. Nonetheless, the changes of $F_{max}$ for a certain component and the ratios between components can be used to investigated the differences among samples qualitatively and quantitively. Details can be seen in Stedmon and Bro (2008) and Murphy et al. (2013). We have also added a brief description of $F_{max}$ in the Supplement (Sec. S1.2).

14. P21 line 9: the picture presented by the authors looks much like lichens to me, more than algae. I also thought that algae only lived in aquatic media (including snow), except when associated with fungi in lichens.

R: We have changed the corresponding text in lines 10-11, page 20 as "We found lichens near these two sample sites (Fig. S3), providing evidence for the latter reason".

15. P22 line 18: I feel that figure 9 is not very informative as such.

R: Fig. 9 has been changed to a box plot as Fig. 7 in the revised manuscript, which exhibits the regional variations of fluorescence indices.

[Figure]

**Figure 7.** HIX (shown in red), BIX (shown in blue) and FI (shown in green) of surface snow samples among regions. The meaning of each part of box is same as that in Fig. 5.

16. P23 line 2-5: Figure S6 should be considered as a replacement for figure 10: it is

more informative, while not being really more clustered. Maybe it could be made even better by reducing slightly the font size of the red equations and drawing dashed red lines, thus using size and line type and color to make stress which correlation is important and which one is rejected.

R: We found that Sec. 3.3 seems not very relate to the aim of this study. Therefore, Sec. 3.3 and Fig. 10 have been deleted. Based on the reviewer's suggestions, we have replotted Fig. S3 as Fig. 8 in the revised manuscript.

[Figure]

**Figure 8.** The linear relationships between intensities of (a) C1 and C2, (b) C1 and C3, (c) C2 and C3. The red dashed lines show the fit of the entire dataset, and the blue solid lines show the fit of data excluded site 67 (shown as markers in red). The corresponding fitting parameters are exhibited in the same color, including the equations, correlation coefficients and p-values.

17. P25 line 20: see my previous comment on nss-K$^+$. how is the authors discussion here sensitive to the objection raised in that previous comment?

R: We have corrected K$^+$ to nss-ndust-K$^+$ following Pio et al. (2007), please see our response to comment 10. The related results have also been changed, such as Fig. 9b and Table 4 in the revised manuscript. We can see that the results varied slightly, and also supported our conclusions.

18. P26: I have a hard time evaluating whether the backtrajectory analysis presented here is relevant. Obviously, fires location from MODIS can be active or not when a

back trajectory passes over it. Here, it seems that any back trajectory can overpass any fire location, and be taken into account in the analysis, even if the backtrajectory overpasses on day n, and the fire was only active from day n+3 to day n+10. This seems to weaken a lot the analysis presented here.

R: We have modified the method. To better combined the backward trajectory and the active fire data, only the fire points that were active during the trajectory calculation period (72 h) were taken into consideration. Then, the trajectories were separated into "passed" and "not passed" groups (red and blue lines in Fig. 10, respectively) to assess the potential influence of biomass burning to the receptor regions.

[Figure]

**Figure 10.** 72-h air mass backward trajectories at 500 m above ground level with the initial positions at representative sites (shown as yellow pentagrams) in each region. Trajectories were calculated four times per day for a period of 30 days preceding the sampling date at a given site by HYSPLIT (version 4, NOAA) except for panel (c). Since the snow was fresh at site 84, the trajectories were derived for 5 days preceding the sampling date. The red lines show the airmasses passed through the active fires before reaching the receptor sites, and the blue lines are those did not pass the fires. The white dots represent the typical industrial cities in Xinjiang, i.e., Karamay, Kuytun, Shihezi and Urumqi from west to east.

**References**

Helms, J. R., Stubbins, A., Ritchie, J. D., Minor, E. C., Kieber, D. J., and Mopper, K.: Absorption spectral slopes and slope ratios as indicators of molecular weight, source, and photobleaching of chromophoric dissolved organic matter, Limnol. Oceanogr., 53, 955-969, 2008.

Murphy, K. R., Stedmon, C. A., Graeber, D., and Bro, R.: Fluorescence spectroscopy and multi-way techniques. PARAFAC, Anal. Methods-Uk., 5, 6557-6566, 2013.

Pio, C. A., Legrand, M., Oliveira, T., Afonso, J., Santos, C., Caseiro, A., Fialho, P., Barata, F., Puxbaum, H., Sanchez-Ochoa, A., Kasper-Giebl, A., Gelencser, A., Preunkert, S., and Schock, M.: Climatology of aerosol composition (organic versus inorganic) at nonurban sites on a west-east transect across Europe, J. Geophys. Res.-Atmos., 112, D23S02, doi:10.1029/2006JD008038, 2007.

Pu, W., Wang, X., Wei, H. L., Zhou, Y., Shi, J. S., Hu, Z. Y., Jin, H. C., and Chen, Q. L.: Properties of black carbon and other insoluble light-absorbing particles in seasonal snow of northwestern China, The Cryosphere, 11, 1213-1233, 2017.

Stedmon, C. A., Markager, S., and Bro, R.: Tracing dissolved organic matter in aquatic environments using a new approach to fluorescence spectroscopy, Mar. Chem., 82, 239-254, 2003.

Stedmon, C. A., and Bro, R.: Characterizing dissolved organic matter fluorescence with parallel factor analysis: a tutorial, Limnol. Oceanogr.-Meth., 6, 572-579, 2008.

Yamashita, Y., Jaffe, R., Maie, N., and Tanoue, E.: Assessing the dynamics of dissolved organic matter (DOM) in coastal environments by excitation emission matrix fluorescence and parallel factor analysis (EEM-PARAFAC), Limnol. Oceanogr., 53, 1900-1908, 2008.

Zhang, Y. L., Zhang, E. L., Yin, Y., van Dijk, M. A., Feng, L. Q., Shi, Z. Q., Liu, M. L., and Qin, B. Q.: Characteristics and sources of chromophoric dissolved organic matter in lakes of the Yungui Plateau, China, differing in trophic state and altitude, Limnol. Oceanogr., 55, 2645-2659, 2010.

Zhao, Y., Song, K., Wen, Z., Li, L., Zang, S., Shao, T., Li, S., and Du, J.: Seasonal characterization of CDOM for lakes in semiarid regions of Northeast China using excitation–emission matrix fluorescence and parallel factor analysis (EEM–PARAFAC), Biogeosciences, 13, 1635-1645, 2016.

---

## Referee Report (RR1)

**Second review of the manuscript number: TC-2018-122**

**Specific comments**
* * *
Page 2: Please use $a_{\mathrm{CDOM}}(280)$ instead of $a(280)$. The latter is often used for total absorption. Please change everywhere in the text.
* * *
Page 3: from the in-situ microbial processes (autochthonous) (Anesio et al., 2009) -> from in-situ processes (autochthonous) such as microbial activity (Anesio et al., 2009)
* * *
Page 5: methods of characterizing -> methods for characterizing
* * *
Page 6: the samples were subjected -> the samples were characterized
* * *
Page 7:

> the spatial variations of CDOM optical properties

Only the spatial variability of CDOM was studied? What about FDOM that was previously discussed?
* * *
Page 8: Although the previous studies -> Although previous studies
* * *
Page 9:

> After PARAFAC analysis, the EEMs can be decomposed into several components with clear chemical interpretations.

This sentence is misleading. The PARAFAC analysis is used to do the decomposition.
* * *
Page 11:

> was determined both by a linear fit and an exponential fit.

Why have you used two fitting approaches?
* * *
Page: 12:

> Finally, linear fit was adopted due to the higher fitting coefficients.

This is not a valid reason to chose the linear approach. Also, do not talk about the non-linear method (and the equation of it) if you are not using it in the paper.
* * *
Page 14:

> Finally, the unweighted average method was chosen due to the highest correlation coefficients.

It is a bit strange to select an approach just because it gives the highest correlation.

Page 16:

> which shows the similar values

should be "which shows  similar values". There are many small errors like this in the manuscript. A careful revision of English writing should be done.

Page 27:

> As presented by Doherty et al. (2013), the mixing ratio of BC in Barrow snow ranged from 10-30 ng g -1 . Hence, the absorption of CDOM in Alaskan snow can be safely ignored, but this does not appear reasonable for some areas across northwestern China.

I do not understand. What is the relational for saying that because BC ranged  between 10-30 ng, CDOM in the Alaskan can be *safely* ignored?

Figure 11: Only one sentence is presenting the result of this figure on page 27. Can you discuss that? Why have you specifically chosen 400 and 500 nm? What could possibly explain the observed differences among sites?

Table 1: Why some observations have a value for a280 but not for the spectral slope?

The authors use *Fig.* and *Figure". Please uniformize in the manuscript.

---

## Author Response (AR2)

Response to reviewer#1

We are very grateful for the reviewer's insightful comments, which are helpful and valuable for greatly improving our manuscript. We have addressed all of the comments carefully as detailed below in our point-by-point responses. Our responses start with "R:".

**Specific comments:**

1. Page 2: Please use $a_{CDOM}(280)$ instead of $a_{280}$. The latter is often used for total absorption. Please change everywhere in the text.

R: All "$a_{280}$" in the manuscript have been changed to "$a_{CDOM}(280)$".

2. Page 3: from the in-situ microbial processes (autochthonous) (Anesio et al., 2009) -> from in-situ processes (autochthonous) such as microbial activity (Anesio et al., 2009).

R: Changed as suggested in lines 16-17, page 3.

3. Page 5: methods of characterizing -> methods for characterizing.

R: Changed as suggested in line 19, page 5.

4. Page 6: the samples were subjected -> the samples were characterized.

R: This sentence has been changed in lines 18-22, page 6, as follows: "In this study, for the first time, with the aim of presenting a comprehensive understanding of CDOM in seasonal snow across northwestern China, UV-vis absorption, fluorescence, and chemical analyses were applied to investigate the abundances, optical properties, and potential sources of CDOM as well as their spatial distributions."

5. Page 7: the spatial variations of CDOM optical properties
   Only the spatial variability of CDOM was studied? What about FDOM that was previously discussed?

R: This sentence has been changed in lines 5-8, page 7, as follows:

"Based on Pu et al. (2017), these sites were separated into five regions by their geographical distribution to investigate the spatial variations of light absorption and fluorescence properties, as well as the potential sources of CDOM."

6. Page 8: Although the previous studies -> Although previous studies.

R: Changed as suggested in line 6, page 8.

7. Page 9: **After PARAFAC analysis**, the EEMs can be decomposed into several components with clear chemical interpretations.

   This sentence is misleading. The PARAFAC analysis is used to do the decomposition.

R: This sentence has been changed in lines 21-22, page 9, as follows:

"PARAFAC analysis can decompose the EEMs into several components with clear chemical interpretations."

8. Page 11: was determined both by a linear fit and an exponential fit.

   Why have you used two fitting approaches?

R: This approach is consistent with Helms et al. (2008), who first introduced $S_{275-295}$ into the CDOM research field. They calculated $S_{275-295}$ using both log-transform linear method and exponential method, and the variation between these methods was less than 1%. Our results showed that this variation was approximately 3% on average. However, for some samples with very low $a_{CDOM}(280)$, the differences of $S_{275-295}$ calculated by linear and exponential regressions were larger than 10%. It might be explained by the higher uncertainties of absorption measurements for samples with low CDOM abundances. Therefore, such values were removed and not considered into further analysis. As discussed above, we used two regression methods for comparing with previous studies and distinguishing the values with high uncertainties.

9. Page: 12: Finally, linear fit was adopted due to the higher fitting coefficients.

   (1) This is not a valid reason to choose the linear approach.

(2) Also, do not talk about the non-linear method (and the equation of it) if you are not using it in the paper.

R: (1) Because regressions were conducted in a narrow wavelength band (275-295 nm), the data points were much fewer than the whole measured wavelengths. As a result, it could lead to a better performance of linear regression than exponential regression, which was proved by the higher $R^2$ of linear method. In addition, the linear regression of log-transferred absorption spectrum has also been frequently used to calculate $S_{275-295}$ in previous studies (Fichot and Benner, 2012; Helms et al., 2008; Yang et al., 2013). Therefore, we chose the linear regression here. The corresponding description has been changed in line 19, page 11 to line 1, page 12, as follows:

"Linear regression has been frequently used to calculate $S_{275-295}$ (Fichot and Benner, 2012; Helms et al., 2008; Yang et al., 2013), and in this study, showed higher $R^2$ values than exponential regression. Therefore, the results of linear regression were adopted here."

(2) The equation of exponential regression has been removed. As mentioned in our response to comment 8, we noted that comparing the results of two methods was helpful to distinguish the values with high uncertainties, hence we suggested that the description of exponential regression could be retained in the manuscript if the reviewer also agreed.

10. Page 14: Finally, the unweighted average method was chosen due to the highest correlation coefficients.

It is a bit strange to select an approach just because it gives the highest correlation.

R: The cophenetic correlation coefficient is a criterion for assessing the efficiency of clustering methods (Saracli et al., 2013). A higher cophenetic correlation coefficient indicates a better clustering method. The sentence in this paragraph has been changed in lines 1-5, page 14, as follows:

"Before determining the clustering method, the cophenetic correlation coefficients, criterions for assessing the efficiency of clustering methods (Saracli et al., 2013), for the cluster trees created by different methods were calculated, including unweighted

average, weighted average, centroid, farthest neighbor, shortest neighbor, weighted center of mass and Ward's methods."

11. Page 16: which shows the similar values

should be "which shows  similar values". There are many small errors like this in the manuscript. A careful revision of English writing should be done.

R: Changed as suggested in lines 13-14, page 16. We have carefully reviewed the manuscript to minimize the grammar errors.

12. Page 27: As presented by Doherty et al. (2013), the mixing ratio of BC in Barrow snow ranged from 10-30 ng g$^{-1}$. Hence, the absorption of CDOM in Alaskan snow can be safely ignored, but this does not appear reasonable for some areas across northwestern China.

I do not understand. What is the relational for saying that because BC ranged  between 10-30 ng, CDOM in the Alaskan can be safely ignored?

R: Sorry for the misleading. We have combined this sentence and the last sentence together to improve the quality of this manuscript. The description has been changed in lines 7-10, page 28, as follows:

"As presented by Doherty et al. (2013), the mixing ratio of BC in Barrow snow ranged between 10-30 ng g$^{-1}$, however, the equivalent BC mixing ratio of CDOM absorption was only 0.14 ng g$^{-1}$ at 400 nm and 0.07 ng g$^{-1}$ at 550 nm (Dang and Hegg, 2014)."

13. Figure 11:

(1) Only one sentence is presenting the result of this figure on page 27. Can you discuss that?

(2) Why have you specifically chosen 400 and 500 nm?

(3) What could possibly explain the observed differences among sites?

R: (1) We have added more detailed discussion of Fig.11 in Sec. 3.4, as follows:

"Most of these sites were assigned to cluster A, except sites 60, 69, and 84. As discussed in Sec. 2.2.2, sites of cluster A exhibited high values of %C1, indicating CDOM mainly

originated from soil dust. At sites 50, 52, and 79, the light absorptions of CDOM and BC were roughly equal at 400 nm. It was not only due to the high abundances of CDOM, but also the relatively low BC mixing ratios in snow (approximately 30 ng g$^{-1}$, Pu et al., 2017). Sites 60, 69, and 84, where the fluorescence intensities were dominated by C2, were the only three sites assigned to cluster C. Biomass burning and anthropogenic pollution (e.g., fossil fuel combustion) are both major sources of fluorophore C2 and BC. Therefore, the BC mixing ratios were approximately 300 ng g$^{-1}$ at these sites (Pu et al., 2017), leading to quite low ratios of light absorption due to CDOM and BC (approximately 0.03 on average at 400 nm). At other sites, this value was typically in the range of 0.1 to 0.4."

(2) In this section, we focused on the influence of CDOM on snow radiation budget rather than snow photochemistry processes which are active within the ultraviolet (UV) wavelengths. The solar radiation in the UV wavelengths is much lower than that in the visible wavelengths (Fig. 4 of Wang et al., 2013), and the observation of snow albedo usually starts form 400 nm (Wang et al., 2017), hence, the absorption at 400 nm were selected. To reflect the wavelength dependences of absorptions for BC and CDOM, we also chose the absorption at 500 nm for comparison, where also the peak value of solar radiation appears.

(3) Please see our response to comment 13(1).

14. Table 1: Why some observations have a value for $a_{280}$ but not for the spectral slope?

R: As mentioned in lines 1-2, page 12 in the revised manuscript, we removed the $S_{275-295}$ values if the difference in $S_{275-295}$ between linear and exponential methods was higher than 10%. Please see details in our response to comment 8.

15. The authors use "Fig." and "Figure". Please uniformize in the manuscript.

R: We have carefully checked the whole manuscript. According to the Author's Guide of The Cryosphere, we used "Figure" when it appeared at the beginning of a sentence, and the abbreviation "Fig." was used in the other situations.

R: Actually, due to the limited space in our vehicle, we had to melt and filtrate the snow samples during the campaign for easy storage. Therefore, we added this paragraph into the manuscript to evaluate the potential influence of freeze-thaw process on optical properties of snow CDOM. According to previous studies, the freeze-thaw process may

influence the relative contributions of PARAFAC components slightly, and the effects on $a_{CDOM}(280)$ and fluorescence indices can be neglected. As discussed in Sec. 3.2.2, if we took the biases due to freezing into consideration, the percentages of PARAFAC components did not change obviously in our study.

Response to editor

Dear Authors

Thank you for your much improved revised version. The reviewers noted a significant improvement in your paper. Some minor changes are still recommended but I foresee that your paper can be accepted once you take these into account. Please address the recommended changes and as usual provide a version in track changes mode. If you provide a revised version within 2 weeks, I will be able to take a final decision before the end-of-year holidays.

Best regards
Florent Domine

We are deeply grateful for the editor's help during the review process. We have carefully revised the manuscript according to the reviewers' comments. If the editor agreed, we would like to add two authors who made contributions to the revision of this manuscript. Jiecan Cui helped to replot several figures, and Yubin Zhou contributed to modify the language.

[revised manuscript text omitted]

where  K is a constant

(from 240related to 550 nm).DOM concentration. The $R^2$ of all the fitsregressions

($S_{275-295}$ and AAE) were higher than 0.9 and most of them were higher than 0.95.

Because the light absorption within the visible wavelengths of some samples were below the detection limit of the spectrometer, 19 of 39 samples were available for the

5   calculation of AAE.

Note that the "left" samples of sites 51b and 58, which showed abnormal absorption and fluorescence spectra compared to other samples, were supposed to be contaminated, and thereby these two samples were not used in the absorption and fluorescence analyses.

10   **2.4 Soluble ions**

The major soluble ions of surface snow water samples were analyzed with an ion chromatograph (Dionex, Sunnyvale, CA, USA) using an AS11 column for the anions $SO_4^{2-}$, $NO_3^-$, $Cl^-$, and $F^-$ and a CS12 column for the cations $Na^+$, $K^+$, $Ca^{2+}$, $Mg^{2+}$, and $NH_4^+$. The soluble ions showed no obvious differences between filtered and

15   unfiltered samples (Pu et al., 2017). According to Pio et al. (2007), the $K^+$ can be separated into three fractions: sea salt (ss), dust and others (the fraction not related to sea salt and mineral dust, nss-ndust). The nss-ndust-$K^+$ is a good makermarker for biomass burning (Pio et al., 2007). The $Ca^{2+}$ concentrations of our samples were mostly largerhigher than that of $Na^+$, leading to much larger mass ratios of $Ca^{2+}/Na^+$ than

20   that in sea water (0.038) (Pio et al., 2007). Therefore, $Ca^{2+}$ is dominated by the dust fraction and not corrected to nss-$Ca^{2+}$ in this study. nss-ndust-$K^+$ is calculated using

the following formulas (Pio et al., 2007):

$$\text{nss-ndust-K}^+ \;=\; \text{K}^+ \;-\; \text{ss-K}^+ \;-\; \text{dust-K}^+, \qquad\qquad (7\underline{6})$$

$$\text{ss-K}^+ \;=\; 0.038 \times \text{ss-Na}^+, \qquad\qquad (8\underline{7})$$

$$\text{ss-Na}^+ \;=\; \text{Na}^+ \;-\; 0.14 \times \text{Ca}^{2+}, \qquad\qquad (9\underline{8})$$

$$\text{dust-K}^+ \;=\; 0.028 \times \text{Ca}^{2+}. \qquad\qquad (1\underline{09})$$

In Eq. ($8\underline{7}$), 0.038 is the mass ratio of $\text{K}^+/\text{Na}^+$ in  sea water (Pio et al., 2007). In Eq. ($9\underline{8}$), the lowest mass ratio of $\text{Na}^+/\text{Ca}^{2+}$ of our samples (0.14) is used to evaluate the dust fraction of $\text{Na}^+$. Similarly, the lowest mass ratio of $\text{K}^+/\text{Ca}^{2+}$ (0.028) is used in Eq. ($1\underline{09}$) to calculate the dust fraction of $\text{
[revised manuscript text omitted]